# Unleashing the Power of Randomization in Auditing Differentially Private ML

**Krishna Pillutla**[1]     **Galen Andrew**[1]     **Peter Kairouz**[1]
**H. Brendan McMahan**[1]     **Alina Oprea**[1,2]     **Sewoong Oh**[1,3]
[1]Google Research          [2]Northeastern University          [3]University of Washington

## Abstract

We present a rigorous methodology for auditing differentially private machine learning algorithms by adding multiple carefully designed examples called canaries. We take a first principles approach based on three key components. First, we introduce Lifted Differential Privacy (LiDP) which expands the definition of differential privacy to handle randomized datasets. This gives us the freedom to design randomized canaries. Second, we audit LiDP by trying to distinguish between the model trained with $K$ canaries versus $K - 1$ canaries in the dataset, leaving one canary out. By drawing the canaries i.i.d., LiDP can leverage the symmetry in the design and reuse each privately trained model to run multiple statistical tests, one for each canary. Third, we introduce novel confidence intervals that take advantage of the multiple test statistics by adapting to the empirical higher-order correlations. Together, this new recipe demonstrates significant improvements in sample complexity, both theoretically and empirically, using synthetic and real data. Further, recent advances in designing stronger canaries can be readily incorporated into the new framework.

## 1   Introduction

Differential privacy (DP), introduced in [22], has gained widespread adoption by governments, companies, and researchers by formally ensuring plausible deniability for participating individuals. This is achieved by guaranteeing that a curious observer of the output of a query cannot be confident in their answer to the following binary hypothesis test: *did a particular individual participate in the dataset or not?* For example, introducing sufficient randomness when training a model on a certain dataset ensures a desired level of differential privacy. This in turn ensures that an individual's sensitive information cannot be inferred from the trained model with high confidence. However, calibrating the right amount of noise can be a challenging process. It is easy to make mistakes in implementing a DP mechanism due to intricacies involving micro-batching, sensitivity analysis, and privacy accounting. Even with correct implementation, there are several incidents of published DP algorithms with miscalculated privacy guarantees that falsely report higher levels of privacy [16, 34, 40, 47, 57, 58]. Data-driven approaches to auditing a mechanism for a violation of a claimed privacy guarantee can significantly mitigate the danger of unintentionally leaking sensitive data.

Popular approaches for auditing privacy share three common components [e.g. 31–33, 45, 69]. Conceptually, these approaches are founded on the definition of DP and involve producing counterexamples that potentially violate the DP condition. Algorithmically, this leads to the standard recipe of injecting a single carefully designed example, referred to as a *canary*, and running a statistical hypothesis test for its presence from the outcome of the mechanism. Analytically, a high-confidence bound on the DP condition is derived by calculating the confidence intervals of the corresponding Bernoulli random variables from $n$ independent trials of the mechanism.

37th Conference on Neural Information Processing Systems (NeurIPS 2023).

Recent advances adopt this standard approach and focus on designing stronger canaries to reduce the number of trials required to successfully audit DP [e.g., 31–33, 39, 45, 69]. However, each independent trial can be as costly as training a model from scratch; refuting a false claim of $(\varepsilon, \delta)$-DP with a minimal number of samples is of utmost importance. In practice, standard auditing can require training on the range of thousands to hundreds of thousands of models [e.g., 60]. Unfortunately, under the standard recipe, we are fundamentally limited by the $1/\sqrt{n}$ sample dependence of the Bernoulli confidence intervals.

**Contributions.** We break this $1/\sqrt{n}$ barrier by rethinking auditing from first principles.

1. **Lifted DP**: We propose to audit an equivalent definition of DP, which we call Lifted DP in §3.1. This gives an auditor the freedom to design a counter-example consisting of *random* datasets and rejection sets. This enables adding random canaries, which is critical in the next step. Theorem 3 shows that violation of Lifted DP implies violation of DP, justifying our framework.

2. **Auditing Lifted DP with Multiple Random Canaries**: We propose adding $K > 1$ canaries under the alternative hypothesis and comparing it against a dataset with $K - 1$ canaries, leaving one canary out. If the canaries are deterministic, we need $K$ separate null hypotheses; each hypothesis leaves one canary out. Our new recipe overcomes this inefficiency in §3.2 by drawing *random* canaries independently from the same distribution. This ensures the exchangeability of the test statistics, allowing us to reuse each privately trained model to run multiple hypothesis tests in a principled manner. This is critical in making the confidence interval sample efficient.

3. **Adaptive Confidence Intervals**: Due to the symmetry of our design, the test statistics follow a special distribution that we call *eXchangeable Bernoulli (XBern)*. Auditing privacy boils down to computing confidence intervals on the average test statistic over the $K$ canaries included in the dataset. If the test statistics are independent, the resulting confidence interval scales as $1/\sqrt{nK}$. However, in practice, the dependence is non-zero and unknown. We propose a new and principled family of confidence intervals in §3.3 that adapts to the empirical higher-order correlations between the test statistics. This gives significantly smaller confidence intervals when the actual dependence is small, both theoretically (Proposition 4) and empirically.

4. **Numerical Results**: We audit an unknown Gaussian mechanism with black-box access and demonstrate (up to) $16\times$ improvement in the sample complexity. We also show how to seamlessly lift recently proposed canary designs in our recipe to improve the sample complexity on real data.

## 2   Background

We describe the standard recipe to audit DP. Formally, we adopt the so-called add/remove definition of differential privacy; our constructions also seamlessly extend to other choices of neighborhoods as we explain in §B.2.

**Definition 1** (Differential privacy). A pair of datasets, $(D_0, D_1)$, is said to be *neighboring* if their sizes differ by one and the datasets differ in one entry: $|D_1 \setminus D_0| + |D_0 \setminus D_1| = 1$. A randomized mechanism $\mathcal{A} : \mathcal{Z}^* \to \mathcal{R}$ is said to be $(\varepsilon, \delta)$-*Differentially Private* (DP) for some $\varepsilon \geq 0$ and $\delta \in [0, 1]$ if it satisfies $\mathbb{P}_{\mathcal{A}}(\mathcal{A}(D_1) \in R) \leq e^{\varepsilon} \, \mathbb{P}_{\mathcal{A}}(\mathcal{A}(D_0) \in R) + \delta$, which is equivalent to

$$\varepsilon \geq \log\left(\mathbb{P}(\mathcal{A}(D_1) \in R) - \delta\right) - \log \mathbb{P}(\mathcal{A}(D_0) \in R), \tag{1}$$

for all pairs of neighboring datasets, $(D_0, D_1)$, and all measurable sets, $R \subset \mathcal{R}$, of the output space $\mathcal{R}$, where $\mathbb{P}_{\mathcal{A}}$ is over the randomness internal to the mechanism $\mathcal{A}$. Here, $\mathcal{Z}^*$ is a space of datasets.

For small $\varepsilon$ and $\delta$, one cannot infer from the output whether a particular individual is in the dataset or not with a high success probability. For a formal connection, we refer to [35]. This naturally leads to a standard procedure for auditing a mechanism $\mathcal{A}$ claiming $(\varepsilon, \delta)$-DP: present $(D_0, D_1, R) \in \mathcal{Z}^* \times \mathcal{Z}^* \times \mathcal{R}$ that violates Eq. (1) as a piece of evidence. Such a counter-example confirms that an adversary attempting to test the participation of an individual will succeed with sufficient probability, thus removing the potential for plausible deniability for the participants.

**Standard Recipe: Adding a Single Canary.** When auditing DP model training (using e.g. DP-SGD [1, 54]), the following recipe is now standard for designing a counter-example $(D_0, D_1, R)$ [31–33, 45, 69]. A training dataset $D_0$ is assumed to be given. This ensures that the model under scrutiny matches the use-case and is called a *null hypothesis*. Next, under a corresponding *alternative hypothesis*, a neighboring dataset $D_1 = D_0 \cup \{c\}$ is constructed by adding a single carefully-designed

example $c \in \mathcal{Z}$, known as a *canary*. Finally, Eq. (1) is evaluated with a choice of $R$ called a *rejection set*. For example, one can reject the null hypothesis (and claim the presence of the canary) if the loss on the canary is smaller than a fixed threshold; $R$ is a set of models satisfying this rejection rule.

**Bernoulli Confidence Intervals.** Once a counter-example $(D_0, D_1, R)$ is selected, we are left to evaluate the DP condition in Eq. (1). Since the two probabilities in the condition cannot be directly evaluated, we rely on the samples of the output from the mechanism, e.g., models trained with DP-SGD. This is equivalent to estimating the expectation, $\mathbb{P}(\mathcal{A}(D) \in R)$ for $D \in \{D_0, D_1\}$, of a Bernoulli random variable, $\mathbb{I}(\mathcal{A}(D) \in R)$, from $n$ i.i.d. samples. Providing high confidence intervals for Bernoulli distributions is a well-studied problem with several off-the-shelf techniques, such as Clopper-Pearson, Jeffreys, Bernstein, and Wilson intervals. Concretely, let $\hat{\mathbb{P}}_n(\mathcal{A}(D) \in R)$ denote the empirical probability of the model falling in the rejection set in $n$ independent runs. The standard intervals scale as $|\mathbb{P}(\mathcal{A}(D_0) \in R) - \hat{\mathbb{P}}_n(\mathcal{A}(D_0) \in R)| \leq C_0 n^{-1/2}$ and $|\mathbb{P}(\mathcal{A}(D_1) \in R) - \hat{\mathbb{P}}_n(\mathcal{A}(D_1) \in R)| \leq C_1 n^{-1/2}$ for constants $C_0$ and $C_1$ independent of $n$. If $\mathcal{A}$ satisfies a claimed $(\varepsilon, \delta)$-DP in Eq. (1), then the following finite-sample lower bound holds with high confidence:

$$\varepsilon \geq \hat{\varepsilon}_n = \log\left(\hat{\mathbb{P}}_n(\mathcal{A}(D_1) \in R) - \frac{C_1}{\sqrt{n}} - \delta\right) - \log\left(\hat{\mathbb{P}}_n(\mathcal{A}(D_0) \in R) + \frac{C_0}{\sqrt{n}}\right). \quad (2)$$

Auditing $(\varepsilon, \delta)$-DP amounts to testing the violation of this condition. This is fundamentally limited by the $n^{-1/2}$ dependence of the Bernoulli confidence intervals. Our goal is to break this barrier.

**Notation.** While the DP condition is symmetric in $(D_0, D_1)$, we use $D_0, D_1$ to refer to specific hypotheses. For symmetry, we need to check both conditions: Eq. (1) and its counterpart with $D_0, D_1$ interchanged. We omit this second condition for notational convenience. We use the shorthand $[k] := \{1, 2, \ldots, k\}$. We refer to random variables by boldfaced letters (e.g. $\boldsymbol{D}$ is a random dataset).

**Related Work.** We provide a detailed survey in Appendix A. A stronger canary (and its rejection set) can increase the RHS of (1). The resulting hypothesis test can tolerate larger confidence intervals, thus requiring fewer samples. This has been the focus of recent breakthroughs in privacy auditing in [32, 39, 41, 45, 60]. They build upon membership inference attacks [e.g. 12, 53, 68] to measure memorization. Our aim is not to innovate on this front. Instead, our framework can seamlessly adopt recently designed canaries and inherit their strengths as demonstrated in §5 and §6.

Random canaries have been used in prior work, but for making the canary out-of-distribution in a computationally efficient manner. No variance reduction is achieved by such random canaries. Adding multiple (deterministic) canaries has been explored in literature but for different purposes. [32, 39] include multiple copies of the same canary to make the canary easier to detect while paying for group privacy since the paired datasets differ in multiple entries (see §3.2 for a detailed discussion).

[42, 69] propose adding multiple distinct canaries to reuse each trained model for multiple hypothesis tests. However, each canary is no stronger than a single canary case, and the resulting auditing suffers from group privacy. When computing the lower bound on $\varepsilon$, however, group privacy is ignored and the test statistics are assumed to be independent without rigorous justification. [2] avoids group privacy in the federated scenario where the adversary has the freedom to return a canary gradient update of choice. The prescribed random gradient shows good empirical performance. The confidence interval is not rigorously derived. Our recipe for injecting multiple canaries *without* a group privacy cost with *rigorous* confidence intervals can be incorporated into these works to give provable lower bounds.

In an independent and concurrent work, Steinke et al. [56] also consider auditing with randomized canaries that are Poisson-sampled, i.e., each canary is included or excluded independently with equal probability. Their recipe involves computing an empirical lower bound by comparing the accuracy (rather than the full confusion matrix as in our case) from the possibly dependent guesses with the worst-case randomized response mechanism. This allows them to use multiple dependent observations from a single trial to give a high probability lower bound on $\varepsilon$. Their confidence intervals, unlike the ones we give here, are non-adaptive and worst-case.

## 3 A New Framework for Auditing DP Mechanisms with Multiple Canaries

We define Lifted DP, a new definition of privacy that is equivalent to DP (§3.1). This allows us to define a new recipe for auditing with multiple random canaries, as opposed to a single deterministic canary in the standard recipe, and reuse each trained model to run multiple correlated hypothesis tests

in a principled manner (§3.2). The resulting test statistics form a vector of *dependent but exchangeable* indicators (which we call an eXchangeable Bernoulli or XBern distribution), as opposed to a single Bernoulli distribution. We leverage this exchangeability to give confidence intervals for the XBern distribution that can potentially improve with the number of injected canaries (§3.3). The pseudocode of our approach is provided in Algorithm 1.

## 3.1 From DP to Lifted DP

To enlarge the design space of counter-examples, we introduce an equivalent definition of DP.

**Definition 2** (Lifted differential privacy)**.** Let $\mathcal{P}$ denote a joint probability distribution over $(\boldsymbol{D}_0, \boldsymbol{D}_1, \boldsymbol{R})$ where $(\boldsymbol{D}_0, \boldsymbol{D}_1) \in \mathcal{Z}^* \times \mathcal{Z}^*$ is a pair of *random* datasets that are neighboring (as in the standard definition of neighborhood in Definition 1) with probability one and let $\boldsymbol{R} \subset \mathcal{R}$ denote a *random* rejection set. We say that a randomized mechanism $\mathcal{A} : \mathcal{Z}^* \to \mathcal{R}$ satisfies $(\varepsilon, \delta)$-*Lifted Differential Privacy* (LiDP) for some $\varepsilon \geq 0$ and $\delta \in [0, 1]$ if, for all $\mathcal{P}$ independent of $\mathcal{A}$, we have

$$\mathbb{P}_{\mathcal{A}, \mathcal{P}}(\mathcal{A}(\boldsymbol{D}_1) \in \boldsymbol{R}) \ \leq \ e^{\varepsilon} \, \mathbb{P}_{\mathcal{A}, \mathcal{P}}(\mathcal{A}(\boldsymbol{D}_0) \in \boldsymbol{R}) + \delta \,. \tag{3}$$

In Appendix A.3, we discuss connections between Lifted DP and other existing extensions of DP, such as Bayesian DP and Pufferfish, that also consider randomized datasets. The following theorem shows that LiDP is equivalent to the standard DP, which justifies our framework of checking the above condition; if a mechanism $\mathcal{A}$ violates the above condition then it violates $(\varepsilon, \delta)$-DP.

**Theorem 3.** *A randomized algorithm $\mathcal{A}$ is $(\varepsilon, \delta)$-LiDP iff $\mathcal{A}$ is $(\varepsilon, \delta)$-DP.*

A proof is provided in Appendix B. In contrast to DP, LiDP involves probabilities over both the internal randomness of the algorithm $\mathcal{A}$ and the distribution $\mathcal{P}$ over $(\boldsymbol{D}_0, \boldsymbol{D}_1, \boldsymbol{R})$. This gives the auditor greater freedom to search over a *lifted* space of joint distributions over the paired datasets and a rejection set; hence the name Lifted DP. Auditing LiDP amounts to constructing a *randomized* (as emphasized by the boldface letters) counter-example $(\boldsymbol{D}_0, \boldsymbol{D}_1, \boldsymbol{R})$ that violates (3) as evidence.

## 3.2 From a Single Deterministic Canary to Multiple Random Canaries

Our strategy is to turn the LiDP condition in Eq. (3) into another condition in Eq. (4) below; this allows the auditor to reuse samples, running multiple hypothesis tests on each sample. This derivation critically relies on our carefully designed recipe that incorporates three crucial features: $(a)$ binary hypothesis tests between pairs of stochastically coupled datasets containing $K$ canaries and $K - 1$ canaries, respectively, for some fixed integer $K$, $(b)$ sampling those canaries i.i.d. from the same distribution, and $(c)$ choice of rejection sets, where each rejection set only depends on a single left-out canary. We introduce the following recipe, also presented in Algorithm 1.

We fix a given training set $D$ and a canary distribution $P_{\text{canary}}$ over $\mathcal{Z}$. This ensures that the model under scrutiny is close to the use case. Under the alternative hypothesis, we train a model on a randomized training dataset $\boldsymbol{D}_1 = D \cup \{\boldsymbol{c}_1, \ldots, \boldsymbol{c}_K\}$, augmented with $K$ random canaries drawn i.i.d. from $P_{\text{canary}}$. Conceptually, this is to be tested against $K$ leave-one-out (LOO) null hypotheses. Under the $k^{\text{th}}$ null hypothesis for each $k \in [K]$, we construct a coupled dataset, $\boldsymbol{D}_{0,k} = D \cup \{\boldsymbol{c}_1, \ldots, \boldsymbol{c}_{k-1}, \boldsymbol{c}_{k+1}, \ldots \boldsymbol{c}_K\}$, with $K - 1$ canaries, leaving the $k^{\text{th}}$ canary out. This coupling of $K - 1$ canaries ensures that $(\boldsymbol{D}_{0,k}, \boldsymbol{D}_1)$ is neighboring with probability one. For each left-out canary, the auditor runs a binary hypothesis test with a choice of a random rejection set $\boldsymbol{R}_k$. We restrict $\boldsymbol{R}_k$ to depend only on the canary $\boldsymbol{c}_k$ that is being tested and not the index $k$. For example, $\boldsymbol{R}_k$ can be the set of models achieving a loss on the canary $\boldsymbol{c}_k$ below a predefined threshold $\tau$.

The goal of this LOO construction is to reuse each trained private model to run multiple tests such that the averaged test statistic has a smaller variance for a *given number of models*. Under the standard definition of DP, one can still use the above LOO construction but with fixed and deterministic canaries. This gives no variance gain because evaluating $\mathbb{P}(\mathcal{A}(D_{0,k}) \in R_k)$ in Eq. (1) or its averaged counterpart $(1/K) \sum_{k=1}^{K} \mathbb{P}(\mathcal{A}(D_{0,k}) \in R_k)$ requires training one model to get one sample from the test statistic $\mathbb{I}(\mathcal{A}(D_{0,k}) \in R_k)$. The key ingredient in *reusing trained models* is randomization.

We build upon the LiDP condition in Eq. (3) by noting that the test statistics are exchangeable for i.i.d. canaries. Specifically, we have for any $k \in [K]$ that $\mathbb{P}(\mathcal{A}(\boldsymbol{D}_{0,k}) \in \boldsymbol{R}_k) = \mathbb{P}(\mathcal{A}(\boldsymbol{D}_{0,K}) \in \boldsymbol{R}_K) = \mathbb{P}(\mathcal{A}(\boldsymbol{D}_{0,K}) \in \boldsymbol{R}'_j)$ for any canary $\boldsymbol{c}'_j$ drawn i.i.d. from $P_{\text{canary}}$ and its corresponding

---

**Algorithm 1** Auditing Lifted DP

---

**Input:** Sample size $n$, number of canaries $K$, number of null tests $m$, DP mechanism $\mathcal{A}$, training set $D$, canary generating distribution $P_{\text{canary}}$, threshold $\tau$, failure probability $\beta$, privacy $\delta \in [0,1]$.

1: **for** $i = 1, \ldots, n$ **do**
2:     Randomly generate $K + m$ canaries $\{\boldsymbol{c}_1, \ldots, \boldsymbol{c}_K, \boldsymbol{c}'_1, \ldots, \boldsymbol{c}'_m\}$ i.i.d. from $P_{\text{canary}}$.
3:     $\boldsymbol{D}_0 \leftarrow D \cup \{\boldsymbol{c}_1, \ldots, \boldsymbol{c}_{K-1}\}, \boldsymbol{D}_1 \leftarrow D \cup \{\boldsymbol{c}_1, \ldots, \boldsymbol{c}_K\}$
4:     Train two models $\boldsymbol{\theta}_0 \leftarrow \mathcal{A}(\boldsymbol{D}_0)$ and $\boldsymbol{\theta}_1 \leftarrow \mathcal{A}(\boldsymbol{D}_1)$
5:     Record test statistics $\boldsymbol{x}^{(i)} \leftarrow \left(\mathbb{I}(f_{\boldsymbol{\theta}_1}(\boldsymbol{c}_k) < \tau)\right)_{k=1}^K$ and $\boldsymbol{y}^{(i)} \leftarrow \left(\mathbb{I}(f_{\boldsymbol{\theta}_0}(\boldsymbol{c}'_j) < \tau)\right)_{j=1}^m$
6: Set $\underline{\boldsymbol{p}}_1 \leftarrow \text{XBernLower}\left(\{\boldsymbol{x}^{(i)}\}_{i \in [n]}, \beta/2\right)$ and $\overline{\boldsymbol{p}}_0 \leftarrow \text{XBernUpper}\left(\{\boldsymbol{y}^{(i)}\}_{i \in [n]}, \beta/2\right)$
7: **Return** $\hat{\varepsilon}_n \leftarrow \log\left((\underline{\boldsymbol{p}}_1 - \delta)/\overline{\boldsymbol{p}}_0\right)$ and a guarantee that $\mathbb{P}(\varepsilon < \hat{\varepsilon}_n) \leq \beta$.

---

rejection set $\boldsymbol{R}'_j$ that are statistically independent of $\boldsymbol{D}_{0,K}$. Therefore, we can rewrite the right side of Eq. (3) using $m$ i.i.d. *test canaries* $\boldsymbol{c}'_1, \ldots, \boldsymbol{c}'_m \sim P_{\text{canary}}$ and a single trained model $\mathcal{A}(\boldsymbol{D}_{0,K})$ as

$$\tfrac{1}{K} \sum_{k=1}^K \mathbb{P}_{\mathcal{A},\mathcal{P}}(\mathcal{A}(\boldsymbol{D}_1) \in \boldsymbol{R}_k) \quad \leq \quad \tfrac{e^\varepsilon}{m} \sum_{j=1}^m \mathbb{P}_{\mathcal{A},\mathcal{P}}(\mathcal{A}(\boldsymbol{D}_{0,K}) \in \boldsymbol{R}'_j) + \delta. \tag{4}$$

Checking this condition is sufficient for auditing LiDP and, via Theorem 3, for auditing DP. For each model trained on $\boldsymbol{D}_1$, we record the test statistics of $K$ (correlated) binary hypothesis tests. This is denoted by a random vector $\boldsymbol{x} = (\mathbb{I}(\mathcal{A}(\boldsymbol{D}_1) \in \boldsymbol{R}_k))_{k=1}^K \in \{0,1\}^K$, where $\boldsymbol{R}_k$ is a rejection set that checks for the presence of the $k^{\text{th}}$ canary. Similar to the standard recipe, we train $n$ models to obtain $n$ i.i.d. samples $\boldsymbol{x}^{(1)}, \ldots, \boldsymbol{x}^{(n)} \in \{0,1\}^K$ to estimate the left side of (4) using the empirical mean:

$$\hat{\boldsymbol{\mu}}_1 := \tfrac{1}{n} \sum_{i=1}^n \tfrac{1}{K} \sum_{k=1}^K \boldsymbol{x}_k^{(i)} \quad \in [0,1], \tag{5}$$

where the subscript one in $\hat{\boldsymbol{\mu}}_1$ denotes that this is the empirical first moment. Ideally, if the $K$ tests are independent, the corresponding confidence interval is smaller by a factor of $\sqrt{K}$. In practice, the $K$ tests are correlated and the size of the confidence interval depends on their correlation. We derive principled confidence intervals that leverage the empirically measured correlations in §3.3. We can define $\boldsymbol{y} \in \{0,1\}^m$ and its mean $\hat{\boldsymbol{\nu}}_1$ analogously for the null hypothesis. We provide pseudocode in Algorithm 1 as an example guideline for applying our recipe to auditing DP training, where $f_\theta(z)$ is the loss evaluated on an example $z$ for a model $\theta$. XBernLower() and XBernUpper() respectively return the lower and upper adaptive confidence intervals from §3.3. We instantiate Algorithm 1 with concrete examples of canary design in §5.

**Our Recipe vs. Multiple Deterministic Canaries.** An alternative with deterministic canaries would be to test between $D_0$ with no canaries and $D_1$ with $K$ canaries such that we can get $K$ samples from a single trained model on $D_0$, one for each of the $K$ test statistics $\{\mathbb{I}(\mathcal{A}(D_0) \in R_k)\}_{k=1}^K$. However, due to the fact that $D_1$ and $D_0$ are now at Hamming distance $K$, this suffers from group privacy; we are required to audit for a much larger privacy leakage of $(K\varepsilon, ((e^{K\varepsilon} - 1)/(e^\varepsilon - 1))\delta)$-DP. Under the (deterministic) LOO construction, this translates into $(1/K) \sum_{k=1}^K \mathbb{P}(\mathcal{A}(D_1) \in R_k) \leq e^{K\varepsilon}(1/K) \sum_{k=1}^K \mathbb{P}(\mathcal{A}(D_0) \in R_k) + ((e^{K\varepsilon} - 1)/(e^\varepsilon - 1))\delta$, where if one canary violates the group privacy condition then the average also violates it. We can reuse a single trained model to get $K$ test statistics in the above condition, but each canary is distinct and not any stronger than the one from $(\varepsilon, \delta)$-DP auditing. One cannot obtain stronger counterexamples without sacrificing the sample gain. For example, we can repeat the same canary $K$ times as proposed in [32]. This makes it easier to detect the canary, making a stronger counter-example, but there is no sample gain as we only get one test statistic per trained model. With deterministic canaries, there is no way to avoid this group privacy cost while our recipe does not incur it.

### 3.3 From Bernoulli Intervals to Higher-Order Exchangeable Bernoulli (XBern) Intervals

The LiDP condition in Eq. (4) critically relies on the canaries being sampled i.i.d. and the rejection set only depending on the corresponding canary. For such a symmetric design, auditing boils down to deriving a Confidence Interval (CI) for a special family of distributions that we call *Exchangeable Bernoulli (XBern)*. We derive the CI for the alternate hypothesis, i.e., the left side of Eq. (4). The CI under the null hypothesis is analogous.

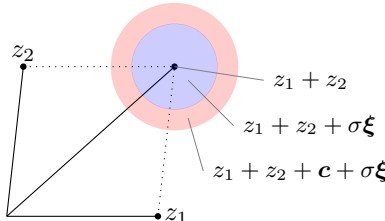

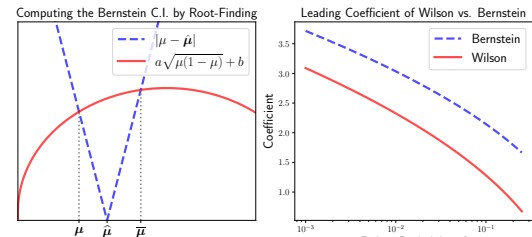

Figure 1: **Bias illustration**: Consider the sum query $z_1 + z_2$ with 2 inputs. Its DP version produces a point in the blue circle w.h.p. due to the noise $\boldsymbol{\xi} \sim \mathcal{N}(0, \boldsymbol{I})$ scaled by $\sigma$. When auditing with a random canary $\boldsymbol{c}$, it contributes additional randomness (red disc) leading to a smaller effective privacy parameter $\varepsilon$.

Figure 2: **Left**: The Bernstein CI $[\underline{\boldsymbol{\mu}}, \overline{\boldsymbol{\mu}}]$ can be found by equating the two sides of Bernstein's inequality in Eq. (6). **Right**: The asymptotic Wilson CI is a tightening of the Bernstein CI with a smaller $1/\sqrt{n}$ coefficient (shown here) and no $1/n$ term.

Recall that $\boldsymbol{x}_k := \mathbb{I}(\mathcal{A}(\boldsymbol{D}_1) \in \boldsymbol{R}_k)$ denotes the test statistic for the $k^{\text{th}}$ canary. By the symmetry of our design, $\boldsymbol{x} \in \{0, 1\}^K$ is an *exchangeable* random vector that is distributed as an exponential family. Further, the distribution of $\boldsymbol{x}$ is fully defined by a $K$-dimensional parameter $(\mu_1, \dots, \mu_K)$ where $\mu_\ell$ is the $\ell^{\text{th}}$ moment of $\boldsymbol{x}$. We call this family XBern. Specifically, this implies permutation invariance of the higher-order moments: $\mathbb{E}[\boldsymbol{x}_{j_1} \cdots \boldsymbol{x}_{j_\ell}] = \mathbb{E}[\boldsymbol{x}_{k_1} \cdots \boldsymbol{x}_{k_\ell}]$ for any distinct sets of indices $(j_l)_{l \in [\ell]}$ and $(k_l)_{l \in [\ell]}$ for any $\ell \le K$. For example, $\mu_1 := \mathbb{E}[(1/K) \sum_{k=1}^K \boldsymbol{x}_k]$, which is the LHS of (4).

Using samples from this XBern, we aim to derive a CI on $\mu_1$ around the empirical mean $\hat{\boldsymbol{\mu}}_1$ in (5). Bernstein's inequality applied to our test statistic $\boldsymbol{m}_1 := (1/K) \sum_{k=1}^K \boldsymbol{x}_k$ gives,

$$|\hat{\boldsymbol{\mu}}_1 - \mu_1| \le \sqrt{\frac{2 \log(2/\beta)}{n} \text{Var}(\boldsymbol{m}_1)} + \frac{2 \log(2/\beta)}{3n}, \tag{6}$$

w.p. at least $1 - \beta$. Bounding $\text{Var}(\boldsymbol{m}_1) \le \mu_1(1 - \mu_1)$ since $\boldsymbol{m}_1 \in [0, 1]$ a.s. and numerically solving the above inequality for $\mu_1$ gives a CI that scales as $1/\sqrt{n}$ — see Figure 2 (left). We call this the **1st-order Bernstein bound**, as it depends only on the 1st moment $\mu_1$. Our strategy is to measure the (higher-order) correlations between $\boldsymbol{x}_k$'s to derive a tighter CI that adapts to the given instance. This idea applies to any standard CI. We derive and analyze the higher order Bernstein intervals here, and experiments use Wilson intervals from Appendix C; see also Figure 2 (right).

Concretely, we can leverage the 2nd order correlation by expanding

$$\text{Var}(\boldsymbol{m}_1) = \tfrac{1}{K}(\mu_1 - \mu_2) + (\mu_2 - \mu_1^2) \quad \text{where} \quad \mu_2 := \tfrac{1}{K(K-1)} \sum_{k_1 < k_2 \in [K]} \mathbb{E}[\boldsymbol{x}_{k_1} \boldsymbol{x}_{k_2}].$$

Ideally, when the second order correlation $\mu_2 - \mu_1^2 = \mathbb{E}[\boldsymbol{x}_1 \boldsymbol{x}_2] - \mathbb{E}[\boldsymbol{x}_1]\mathbb{E}[\boldsymbol{x}_2]$ equals 0, we have $\text{Var}(\boldsymbol{m}_1) = \mu_1(1 - \mu_1)/K$, a factor of $K$ improvement over the worst-case. Our higher-order CIs adapt to the *actual level of correlation* of $\boldsymbol{x}$ by further estimating the 2nd moment $\mu_2$ from samples. Let $\overline{\boldsymbol{\mu}}_2$ be the first-order Bernstein upper bound on $\mu_2$ such that $\mathbb{P}(\mu_2 \le \overline{\boldsymbol{\mu}}_2) \ge 1 - \beta$. On this event,

$$\text{Var}(\boldsymbol{m}_1) \le \tfrac{1}{K}(\mu_1 - \overline{\boldsymbol{\mu}}_2) + (\overline{\boldsymbol{\mu}}_2 - \mu_1^2). \tag{7}$$

Combining this with (6) gives us the **2nd-order Bernstein bound** on $\mu_1$, valid w.p. $1 - 2\beta$. Since $\overline{\boldsymbol{\mu}}_2 \lesssim \hat{\boldsymbol{\mu}}_2 + 1/\sqrt{n}$ where $\hat{\boldsymbol{\mu}}_2$ is the empirical estimate of $\mu_2$, the 2nd-order bound scales as

$$|\mu_1 - \hat{\boldsymbol{\mu}}_1| \lesssim \sqrt{\frac{1}{nK}} + \sqrt{\frac{1}{n}|\hat{\boldsymbol{\mu}}_2 - \hat{\boldsymbol{\mu}}_1^2|} + \frac{1}{n^{3/4}}, \tag{8}$$

where constants and log factors are omitted. Thus, our 2nd-order CI can be as small as $1/\sqrt{nK} + 1/n^{3/4}$ (when $\hat{\boldsymbol{\mu}}_1^2 \approx \boldsymbol{\mu}_2$) or as large as $1/\sqrt{n}$ (in the worst-case). With small enough correlations of $|\hat{\boldsymbol{\mu}}_2 - \hat{\boldsymbol{\mu}}_1^2| = O(1/K)$, this suggests a choice of $K = O(\sqrt{n})$ to get CI of $1/n^{3/4}$. In practice, the correlation is controlled by the design of the canary. For the Gaussian mechanism with random canaries, the correlation indeed empirically decays as $1/K$, as we see in §4. Thus, the CI decreases monotonically with $K$. This is also true for the CI under the null hypothesis with $K$ replaced by $m$.

However, there is a qualitative difference between the null and alternate hypotheses. Estimation under the alternate hypothesis incurs a larger bias as the number $K$ of canaries increases — this is due

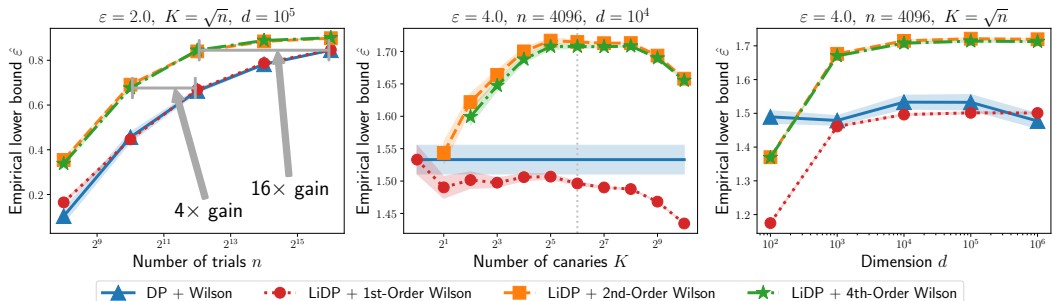

**Figure 3:** **Left**: For Gaussian mechanisms, the proposed LiDP-based auditing with $K$ canaries provides a significant gain in the required number of trials to achieve a desired level of lower bound $\hat{\varepsilon}$ on the privacy. **Center**: Increasing the number of canaries trades off the bias and the variance, with our prescribed $K = \sqrt{n}$ achieving a good performance. **Right**: Increasing the dimension makes the canaries less correlated, thus achieving smaller confidence intervals, and larger $\hat{\varepsilon}$. We shade the standard error over 25 repetitions.

to the additional randomness from adding more canaries into the training process, as illustrated in Figure 1. The optimal choice of $K$ balances the bias and variance. On the other hand, estimation under the null hypothesis incurs no bias and a larger number $m$ of test canaries always helps. We return to the bias-variance tradeoff in §4.

**Higher-order intervals.** We can recursively apply this method by expanding the variance of higher-order statistics, and derive higher-order Bernstein bounds. The next recursion uses empirical $3^{\text{rd}}$ and $4^{\text{th}}$ moments $\hat{\mu}_3$ and $\hat{\mu}_4$ to get the **$4^{\text{th}}$-order Bernstein bound** which scales as

$$|\mu_1 - \hat{\mu}_1| \lesssim \sqrt{\frac{1}{nK}} + \sqrt{\frac{1}{n}|\hat{\mu}_2 - \hat{\mu}_1^2|} + \frac{1}{n^{3/4}}|\hat{\mu}_4 - \hat{\mu}_2^2|^{1/4} + \frac{1}{n^{7/8}} . \tag{9}$$

Ideally, when the $4^{\text{th}}$-order correlation is small enough, $|\hat{\mu}_4 - \hat{\mu}_2^2| = O(1/K)$, (along with the 2nd-order correlation, $\hat{\mu}_2 - \hat{\mu}_1^2$) this 4th-order CI scales as $1/n^{7/8}$ with a choice of $K = O(n^{3/4})$ improving upon the 2nd-order CI of $1/n^{3/4}$. We can recursively derive even higher-order CIs, but we find in §4 that the gains diminish rapidly. In general, the $\ell^{\text{th}}$ order Bernstein bounds achieve CIs scaling as $1/n^{(2\ell-1)/2\ell}$ with a choice of $K = O(n^{(\ell-1)/\ell})$. This shows that the higher-order CI decreases in the order $\ell$ of the correlations used; we refer to Appendix C for details.

**Proposition 4.** *For any positive integer $\ell$ that is a power of two and $K = \lceil n^{(\ell-1)/\ell} \rceil$, suppose we have $n$ samples from a $K$-dimensional XBern distribution with parameters $(\mu_1, \ldots, \mu_K)$. If all $\ell'^{th}$-order correlations scale as $1/K$, i.e., $|\mu_{2\ell'} - \mu_{\ell'}^2| = O(1/K)$, for all $\ell' \leq \ell$ and $\ell'$ is a power of two, then the $\ell^{th}$-order Bernstein bound is $|\mu_1 - \hat{\mu}_1| = O(1/n^{(2\ell-1)/(2\ell)})$.*

## 4 Simulations: Auditing the Gaussian Mechanism

**Setup.** We consider a simple sum query $q(D) = \sum_{z \in D} z$ over the unit sphere $\mathcal{Z} = \{z \in \mathbb{R}^d : \|z\|_2 = 1\}$. We want to audit a Gaussian mechanism that returns $q(D) + \sigma\boldsymbol{\xi}$ with standard Gaussian $\boldsymbol{\xi} \sim \mathcal{N}(0, \boldsymbol{I}_d)$ and $\sigma$ calibrated to ensure $(\varepsilon, \delta)$-DP. We assume black-box access, where we do not know what mechanism we are auditing and we only access it through samples of the outcomes. A white-box audit is discussed in §A.1. We apply our new recipe with canaries sampled uniformly at random from $\mathcal{Z}$. Following standard methods [e.g. 23], we declare that a canary $\boldsymbol{c}_k$ is present if $\boldsymbol{c}_k^\top \mathcal{A}(\boldsymbol{D}) > \tau$ for a threshold $\tau$ learned from separate samples. For more details and additional results, see Appendix E. A broad range of values of $K$ (between 32 and 256 in Figure 3 middle) leads to good performance in auditing LiDP. We use $K = \sqrt{n}$ (as suggested by our analysis in (8)), $m = K$ test canaries, and the $2^{\text{nd}}$-order Wilson estimator (as gains diminish rapidly afterward) as a reliable default setting.

**Sample Complexity Gains.** In Figure 3 (left), the proposed approach of injecting $K$ canaries with the $2^{\text{nd}}$ order Wilson interval (denoted "LiDP +$2^{\text{nd}}$-Order Wilson") reduces the number of trials, $n$, needed to reach the same empirical lower bound, $\hat{\varepsilon}$, by $4\times$ to $16\times$, compared to the baseline of injecting a single canary (denoted "DP+Wilson"). We achieve $\hat{\varepsilon}_n = 0.85$ with $n = 4096$ (while the baseline requires $n = 65536$) and $\hat{\varepsilon}_n = 0.67$ with $n = 1024$ (while the baseline requires $n = 4096$).

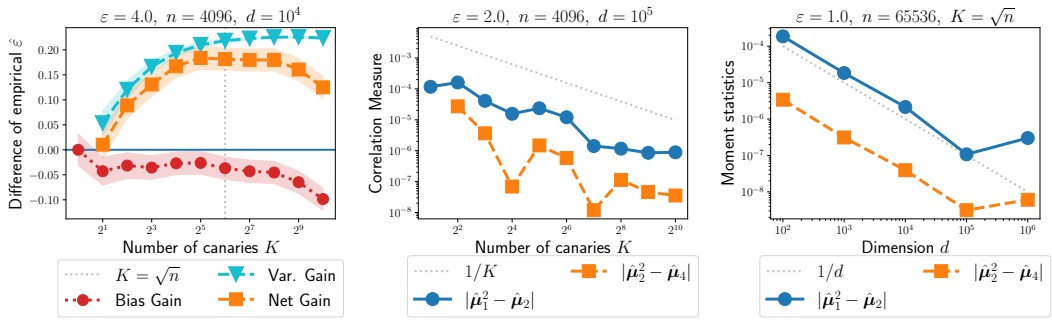

Figure 4: **Left**: Separating the effects of bias and variance in auditing LiDP; cf. definition (10). **Center & Right**: The correlations between the test statistics of the canaries decrease with $K$ and $d$, achieving smaller CIs.

**Number of Canaries and Bias-Variance Tradeoffs.** In Fig. 3 (middle), LiDP auditing with 2nd/4th-order CIs improve with increasing canaries up to a point and then decreases. This is due to a bias-variance tradeoff, which we investigate further in Figure 4 (left). Let $\hat{\varepsilon}(K, \ell)$ denote the empirical privacy lower bound with $K$ canaries using an $\ell^{\text{th}}$ order interval, e.g., the baseline is $\hat{\varepsilon}(1, 1)$. Let the bias from injecting $K$ canaries and the variance gain from $\ell^{\text{th}}$-order interval respectively be

$$\Delta\text{Bias}(K) := \hat{\varepsilon}(K, 1) - \hat{\varepsilon}(1, 1), \quad \text{and} \quad \Delta\text{Var}(K, \ell) := \hat{\varepsilon}(K, \ell) - \hat{\varepsilon}(K, 1). \tag{10}$$

In Figure 4 (left), the gain $\Delta\text{Bias}(K)$ from bias is negative and gets worse with increasing $K$; when testing for each canary, the $K-1$ other canaries introduce more randomness that makes the test more private and hence lowers the $\hat{\varepsilon}$ (see also Figure 1. The gain $\Delta\text{Var}(K)$ in variance is positive and increases with $K$ before saturating. This improved "variance" of the estimate is a key benefit of our framework. The net improvement $\hat{\varepsilon}(K, \ell) - \hat{\varepsilon}(1, 1)$ is a sum of these two effects. This trade-off between bias and variance explains the concave shape of $\hat{\varepsilon}$ in $K$.

Next, we see from Figure 5 that a larger number $m$ of test canaries always helps, i.e., it does not incur a similar bias-variance tradeoff. Indeed, this is because larger $m$ does not lead to any additional bias, as discussed in §3.3. However, the gains quickly saturate, so $m = K$ is a reliable default, especially for $K \geq 16$ or so; we refer to Appendix E for more plots and details.

**Correlation between Canaries.** Based on (8), this improvement in the variance can further be examined by looking at the term $|\hat{\boldsymbol{\mu}}_2 - \hat{\boldsymbol{\mu}}_1^2|$ that leads to a narrower 2nd-order Wilson interval. The log-log plot of this term in Figure 4 (middle) is nearly parallel to the dotted $1/K$ line (slope = $-0.93$), meaning that it decays roughly as $1/K$ (note that the log-log plot of $y = cx^a$ is a straight line with slope $a$). This indicates that we get close to a $1/\sqrt{nK}$ confidence interval as desired. Similarly, we get that $|\hat{\boldsymbol{\mu}}_4 - \hat{\boldsymbol{\mu}}_2^2|$ decays roughly as $1/K$ (slope = $-1.05$). However, the 4th-order estimator offers only marginal additional improvements in the small $\varepsilon$ regime (see Appendix E). Thus, the gain diminishes rapidly in the order of our estimators.

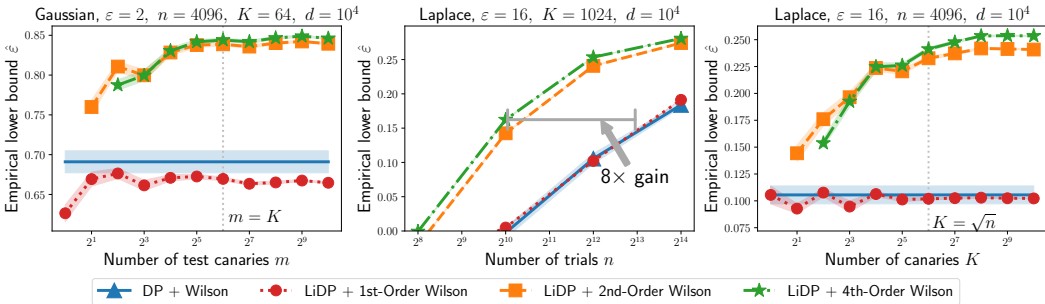

Figure 5: **Left**: There is no bias-variance tradeoff with the number $m$ of test canaries: larger $m$ always gives a better empirical $\hat{\varepsilon}$ but the results saturate quickly. This experiment is for the Gaussian mechanism. **Center & Right**: The proposed LiDP-based auditing procedure gives significant improvements similar to Figure 3 when auditing the Laplace mechanism with canaries sampled randomly from the unit $\ell_1$ ball.

**Effect of Dimension on the Bias and Variance.** As the dimension $d$ increases, the LiDP-based lower bound becomes tighter monotonically as we show in Figure 3 (right). This is due to both the bias gain, as in (10), improving (less negative) and the variance gain improving (more positive). With increasing $d$, the variance of our estimate reduces because the canaries become less correlated. Guided by Eq. (8,9), we measure the relevant correlation measures $|\hat{\boldsymbol{\mu}}_2 - \hat{\boldsymbol{\mu}}_1^2|$ and $|\hat{\boldsymbol{\mu}}_4 - \hat{\boldsymbol{\mu}}_2^2|$ in Figure 4 (right). Both decay approximate as $1/d$ (slope $= -1.06$ and $-1.00$ respectively, ignoring the outlier at $d = 10^6$). This suggests that, for Gaussian mechanisms, the corresponding XBern distribution resulting from our recipe behaves favorably as $d$ increases.

**Auditing Other Mechanisms.** The proposed LiDP-based auditing recipe is agnostic to the actual privacy mechanism. We audit a Laplace mechanism with canaries drawn uniformly at random over the unit $\ell_1$ ball in Figure 4 (center and right). The results are qualitatively similar to those of the Gaussian mechanism, leading to an eight-fold improvement in the sample complexity over the baseline.

## 5   Lifting Existing Canary Designs

Several prior works [e.g., 32, 41, 45, 60], focus on designing stronger canaries to improve the lower bound on $\varepsilon$. We provide two concrete examples of how to *lift* these canary designs to be compatible with our framework while inheriting their strengths. We refer to Appendix D for further details.

We impose two criteria on the distribution $P_{\text{canary}}$ over canaries for auditing LiDP. First, the injected canaries are easy to detect, so that the probabilities on the left side of (4) are large and those on the right side are small. Second, a canary $\boldsymbol{c} \sim P_{\text{canary}}$, if included in the training of a model $\theta$, is unlikely to change the membership of $\theta \in R_{\boldsymbol{c}'}$ for an independent canary $\boldsymbol{c}' \sim P_{\text{canary}}$. Existing canary designs already impose the first condition to audit DP using (1). The second condition ensures that the canaries are uncorrelated, allowing our adaptive CIs to be smaller, as we discussed in §3.3.

**Data Poisoning via Tail Singular Vectors.** ClipBKD [32] adds as canaries the tail singular vector of the input data (e.g., images). This ensures that the canary is out of distribution, allowing for easy detection. We lift ClipBKD by defining the distribution $P_{\text{canary}}$ as the uniform distribution over the $p$ tail singular vectors of the input data. If $p$ is small relative to the dimension $d$ of the data, then they are still out of distribution, and hence, easy to detect. For the second condition, the orthogonality of the singular vectors ensures the interaction between canaries is minimal, as measured empirically.

**Random Gradients.** The approach of [2] samples random vectors (of the right norm) as canary gradients, assuming a grey-box access where we can inject gradients. Since random vectors are nearly orthogonal to any fixed vector in high dimensions, their presence is easy to detect with a dot product. Similarly, any two i.i.d. canary gradients are roughly orthogonal, leading to minimal interactions.

## 6   Experiments

We compare the proposed LiDP auditing recipe relative to the standard one for DP training of machine learning models. Our code is available online.[1]

**Setup.** We test with two classification tasks: FMNIST [65] is a 10-class grayscale image classification dataset, while Purchase-100 is a sparse tabular dataset with 600 binary features and 100 classes [19, 53]. We train a linear model and a multi-layer perceptron (MLP) with 2 hidden layers using DP-SGD [1] to achieve $(\varepsilon, 10^{-5})$-DP with varying values of $\varepsilon$. The training is performed using cross-entropy for a fixed epoch budget and a batch size of 100. We refer to Appendix F for details.

**Auditing.** We audit the LiDP using the two types of canaries from §5: data poisoning and random gradient canaries. We vary the number $K$ of canaries and the number $n$ of trials. We track the empirical lower bound obtained from the Wilson family of confidence intervals. We compare this with auditing DP, which coincides with auditing LiDP with $K = 1$ canary. We audit *only the final model* in all the experiments.

**Sample Complexity Gain.** Table 1 shows the reduction in the sample complexity from auditing LiDP. For each canary type, auditing LiDP is better 11 out of the 12 settings considered. The average improvement (i.e., the harmonic mean over the table) for data poisoning is $2.3\times$, while for random

---

[1] https://github.com/google-research/federated/tree/master/lidp_auditing

| Dataset / Model | C.I./ (Wilson) | Data Poisoning Canary | | | | Random Gradient Canary | | | |
|---|---|---|---|---|---|---|---|---|---|
| | | $\varepsilon = 2$ | $\varepsilon = 4$ | $\varepsilon = 8$ | $\varepsilon = 16$ | $\varepsilon = 2$ | $\varepsilon = 4$ | $\varepsilon = 8$ | $\varepsilon = 16$ |
| **FMNIST / Linear** | 2nd-Ord. | 0.68 | 3.31 | 2.55 | 4.38 | 2.69 | 3.34 | 5.98 | 5.06 |
| | 4th-Ord. | 0.42 | 2.68 | 2.29 | 3.76 | 2.66 | 2.70 | 7.75 | 4.19 |
| **FMNIST / MLP** | 2nd-Ord. | 4.80 | 1.46 | 2.95 | 1.60 | 4.62 | 2.88 | 2.33 | 5.46 |
| | 4th-Ord. | 4.42 | 2.49 | 2.37 | 1.30 | 3.95 | 2.81 | 2.02 | 4.60 |
| **Purchase / MLP** | 2nd-Ord. | 3.14 | 1.06 | 1.41 | 9.28 | 1.41 | 0.71 | 4.30 | 2.84 |
| | 4th-Ord. | 2.84 | 1.09 | 1.36 | 6.99 | 1.29 | 0.42 | 4.35 | 2.38 |

Table 1: The (multiplicative) improvement in the sample complexity from auditing LiDP with $K = 16$ canaries compared to auditing DP with $n = 1000$ trials. We determine this factor by linearly interpolating/extrapolating $\hat{\varepsilon}_n$; cf. Figure 6 (left) for a visual representation of these numbers. For instance, an improvement of 3.31 means LiDP needs $n \approx 1000/3.31 \approx 302$ trials to reach the same empirical lower bound that DP reaches at $n = 1000$.

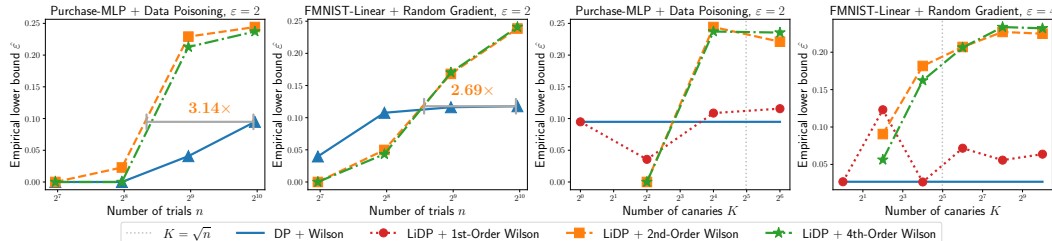

Figure 6: **Left two**: LiDP-based auditing with $K > 1$ canaries achieves the same lower bound $\hat{\varepsilon}$ on the privacy loss with fewer trials. **Right two**: LiDP auditing is robust to $K$; the prescribed $K = \sqrt{n}$ is a reliable default.

gradients, it is $3.0\times$. Since each trial is a full model training run, this improvement can be quite significant in practice. This improvement can also be seen visually in Figure 6 (left two).

**Number of Canaries.** We see from Figure 6 (right two) that LiDP auditing on real data behaves similarly to Figure 3 in §4. We also observe the bias-variance tradeoff in the case of data poisoning (center right). The choice $K = \sqrt{n}$ is competitive with the best value of $K$, validating our heuristic. Overall, these results show that the insights from §4 hold even with the significantly more complicated DP mechanism involved in training models privately.

**Additional Comparisons.** We show in §F.4 that our LiDP auditing with generally outperforms the Bayesian auditing approach of Zanella-Béguelin et al. [69]. This also suggests an interesting future research direction: adapt Bayesian credible intervals for LiDP auditing to get the best of both worlds.

# 7 Conclusion

We introduce a new framework for auditing differentially private learning. Diverging from the standard practice of adding a single deterministic canary, we propose a new recipe of adding multiple i.i.d. random canaries. This is made rigorous by an expanded definition of privacy that we call LiDP. We provide novel higher-order confidence intervals that can automatically adapt to the level of correlation in the data. We empirically demonstrate that there is a potentially significant gain in sample dependence of the confidence intervals, achieving a favorable bias-variance tradeoff.

Although any rigorous statistical auditing approach can benefit from our framework, it is not yet clear how other popular approaches [e.g. 12] for measuring memorization can be improved with randomization. Bridging this gap is an important practical direction for future research. It is also worth considering how our approach can be adapted to audit the diverse definitions of privacy in machine learning [e.g. 28].

**Broader Impact.** Auditing private training involves a trade-off between the computational cost and the tightness of the guarantee. This may not be appropriate for all practical settings. For deployment in production, it is worth further studying approaches with minimal computational overhead [e.g. 2, 12].

**Acknowledgements**

We acknowledge Lang Liu for helpful discussions regarding hypothesis testing and conditional probability, Matthew Jagielski for help in debugging the experiments with data poisoning canaries, and Zeba Islam for help in simplifying proofs involving combinatorics.

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

# Appendix

## Table of Contents

# A  Related Work

Prior to [17], privacy auditing required some access to the description of the mechanism. [5–7, 24, 43, 50, 51, 62] provide platforms with a specific set of functions to be used to implement the mechanism, where the end-to-end privacy of the source code can be automatically verified. Closest to our setting is the work of [18], where statistical testing was first proposed for privacy auditing, given access to an oracle that returns the exact probability measure of the mechanism. However, the guarantee is only for a relaxed notion of DP from [8] and the run-time depends super-linearly on the size of the output domain.

The pioneering work of [17] was the first to propose practical methods to audit privacy claims given a black-box access to a mechanism. The focus was on simple queries that do not involve any training of models. This is motivated by [16, 40], where even simple mechanisms falsely reported privacy guarantees. For such mechanisms, sampling a large number, say $500,000$ in [17], of outputs is computationally easy, and no effort was made in [17] to improve the statistical trade-off. However, for private learning algorithms, training such a large number of models is computationally prohibitive. The main focus of our framework is to improve this statistical trade-off for auditing privacy. We first survey recent breakthroughs in designing stronger canaries, which is orthogonal to our main focus.

## A.1  Auditing Private Machine Learning with Strong Canaries

Recent breakthroughs in auditing are accelerated by advances in privacy attacks, in particular membership inference. An attacker performing membership inference would like to determine if a particular data sample was part of the training set. Early work on membership inference [11, 23, 30, 52] considered algorithms for statistical methods, and more recent work demonstrates black-box attacks on ML algorithms [14, 33, 53, 67, 68]. [33] compares different notions of privacy by measuring privacy empirically for the composition of differential privacy [35], concentrated differential privacy [10, 21], and Rényi differential privacy [44]. This is motivated by [49], which compares different DP mechanisms by measuring the success rates of membership inference attacks. [31] attempts to measure the intrinsic privacy of stochastic gradient descent (without additional noise as in DP-SGD) using canary designs from membership inference attacks. Membership inference attacks have been shown to have higher success when the adversary can poison the training dataset [59].

A more devastating privacy attack is the extraction or reconstruction of the training data, which is particularly relevant for generative models such as large language models (LLMs). Several papers showed that LLMs tend to memorize their training data [13, 15], allowing an adversary to prompt the generative models and extract samples of the training set. The connection between the ability of an attacker to perform training data reconstruction and DP guarantees has been shown in recent work [26, 29].

When performing privacy auditing, a stronger canary design increases the success of the adversary in the distinguishing test and improves the empirical privacy bounds. The resulting hypothesis test can tolerate larger confidence intervals and requires less number of samples. Recent advances in privacy auditing have focused on designing such stronger canaries. [32] designs data poisoning canaries, in the direction of the lowest variance of the training data. This makes the canary out of distribution, making it easier to detect. [45] proposes attack surfaces of varying capabilities. For example, a gradient attack canary returns a gradient of choice when accessed by DP-SGD. It is shown that, with more powerful attacks, the canaries become stronger and the lower bounds become higher. [60] proposes using an example from the baseline training dataset, after changing the label, and introduces a search procedure to find a strong canary. More recently, [46] proposes a significantly improved auditing scheme for DP-SGD under a white-box access model where $(i)$ the auditor knows that the underlying mechanism is DP-SGD with a spherical Gaussian noise with unknown variance, and $(ii)$ all intermediate models are revealed. Since each coordinate of the model update provides an independent sample from the same Gaussian distribution, sample complexity is dramatically improved. [41] proposes CANIFE, a novel canary design method that finds a strong data poisoning canary adaptively under the federated learning scenario.

Prior work in this space shows that privacy auditing can be performed via privacy attacks, such as membership inference or reconstruction, and strong canary design results in better empirical privacy bounds. We emphasize that our aim is not to innovate on optimizing canary design for performing

privacy auditing. Instead, our framework can seamlessly adopt recently designed canaries and inherit their strengths as demonstrated in §5 and §6.

## A.2 Improving Statistical Trade-offs in Auditing

Eq. (2) points to two orthogonal directions that can potentially improve the sample dependence: designing stronger canaries and improving the sample dependence of the confidence intervals. The former was addressed in the previous section. There is relatively less work in improving the statistical dependence, which our framework focuses on.

Given a pair of neighboring datasets, $(D_0, D_1)$, and for a query with discrete output in a finite space, the statistical trade-off of estimating privacy parameters was studied in [25] where a plug-in estimator is shown to achieve an error of $O(\sqrt{de^{2\varepsilon}/n})$, where $d$ is the size of the discrete output space. [38] proposes a sub-linear sample complexity algorithm that achieves an error scaling as $\sqrt{de^{\varepsilon}/n \log n}$, based on polynomial approximation of a carefully chosen degree to optimally trade-off bias and variance motivated by [27, 64]. Similarly, [37] provides a lower bound for auditing Rényi differential privacy. [9] trains a classifier for the binary hypothesis test and uses the classifier to design rejection sets. [3] proposes local search to find the rejection set efficiently. More recently, [69] proposes numerical integration over a larger space of false positive rate and true positive rate to achieve better sample complexity of the confidence region in the two-dimensional space. Our framework can be potentially applied to this confidence region scenario, which is an important future research direction.

## A.3 Connections to Other Notions of Differential Privacy

Similar to Lifted DP, a line of prior works [36, 55, 61, 66] generalizes DP to include a distributional assumption over the dataset. However, unlike Lifted DP, they are motivated by the observation that DP is not sufficient for preserving privacy when the samples are highly correlated. For example, upon releasing the number of people infected with a highly contagious flu within a tight-knit community, the usual Laplace mechanism (with sensitivity one) is not sufficient to conceal the likelihood of one member getting infected when the private count is significantly high. One could apply group differential privacy to hide the contagion of the entire population, but this will suffer from excessive noise. Ideally, we want to add a noise proportional to the *expected* size of the infection. Pufferfish privacy, introduced in [36], achieves this with a generalization of DP that takes into account prior knowledge of a class of potential distributions over the dataset. A special case of $\varepsilon$-Pufferfish with a specific choice of parameters recovers a special case of our Lifted DP in Eq. (3) with $\delta = 0$ [36, Section 3.2], where a special case of Theorem 3 has been proven for pure DP [36, Theorem 3.1]. However, we want to emphasize that our Lifted DP is motivated by a completely different problem of auditing differential privacy and is critical to breaking the barriers in the sample complexity.

# B Properties of Lifted DP and Further Details

## B.1 Equivalence Between DP and LiDP

In contrast to the usual $(\varepsilon, \delta)$-DP in Eq. (1), the probability in LiDP is over both the internal randomness of the algorithm $\mathcal{A}$ and the distribution $\mathcal{P}$ over the triplet $(\boldsymbol{D}_0, \boldsymbol{D}_1, \boldsymbol{R})$. Since we require that the definition holds only for lifted distributions $\mathcal{P}$ that are independent of the algorithm $\mathcal{A}$, it is easy to show its equivalence to the usual notion of $(\varepsilon, \delta)$-DP.

**Theorem 3.** *A randomized algorithm $\mathcal{A}$ is $(\varepsilon, \delta)$-LiDP iff $\mathcal{A}$ is $(\varepsilon, \delta)$-DP.*

*Proof.* Suppose $\mathcal{A}$ is $(\varepsilon, \delta)$-LiDP. Fix a pair of neighboring datasets $D_0, D_1$ and an outcome $R \subset \mathcal{R}$. Define $\mathcal{P}_{D_0, D_1, R}$ as the point mass on $(D_0, D_1, R)$, i.e.,

$$\mathrm{d}\mathcal{P}_{D_0, D_1, R}(D_0', D_1', R') = \mathbb{1}(D_0' = D_0, \, D_1' = D_1, \, R' = R) \,,$$

so that $\mathbb{P}_{\mathcal{A}, \mathcal{P}_{D_0, D_1, R}}(\mathcal{A}(\boldsymbol{D}_0) \in \boldsymbol{R}) = \mathbb{P}(\mathcal{A}(D_0) \in R)$ and similarly for $D_1$. Then, applying the definition of $(\varepsilon, \delta)$-LiDP w.r.t. the distribution $\mathcal{P}_{D_0, D_1, R}$ gives

$$\mathbb{P}_{\mathcal{A}}(\mathcal{A}(D_1) \in R) \leq e^{\varepsilon} \, \mathbb{P}_{\mathcal{A}}(\mathcal{A}(D_0) \in R) + \delta \,.$$

Since this holds for any neighboring datasets $D_0, D_1$ and outcome set $R$, we get that $\mathcal{A}$ is $(\varepsilon, \delta)$-DP.

Conversely, suppose that $\mathcal{A}$ is $(\varepsilon, \delta)$-DP. For any distribution $\mathcal{P}$ over pairs of neighboring datasets $D_0, D_1$ and outcome set $R$, we have by integrating the DP definition

$$\int \mathbb{P}(\mathcal{A}(D_1) \in R) \, d\mathcal{P}(D_0, D_1, R) \leq e^{\varepsilon} \int \mathbb{P}(\mathcal{A}(D_0) \in R) \, d\mathcal{P}(D_0, D_1, R) + \delta. \qquad (11)$$

Next, we use the law of iterated expectation to get

$$
\begin{aligned}
\mathbb{P}_{\mathcal{A},\mathcal{P}}\big(\mathcal{A}(\boldsymbol{D}_0) \in \boldsymbol{R})\big) &= \mathbb{E}_{\mathcal{A},\mathcal{P}}\left[\mathbb{I}\big(\mathcal{A}(\boldsymbol{D}_0) \in \boldsymbol{R}\big)\right] \\
&= \mathbb{E}_{(\boldsymbol{D}_0, \boldsymbol{D}_1, \boldsymbol{R}) \sim \mathcal{P}}\left[\mathbb{E}_{\mathcal{A}}\left[\mathbb{I}\big(\mathcal{A}(\boldsymbol{D}_0) \in \boldsymbol{R}\big) \,\big|\, \boldsymbol{D}_0, \boldsymbol{R}\right]\right] \\
&= \int \mathbb{E}_{\mathcal{A}}\left[\mathbb{I}\big(\mathcal{A}(\boldsymbol{D}_0) \in \boldsymbol{R}\big) \,\big|\, \boldsymbol{D}_0 = D_0, \boldsymbol{R} = R\right] \, d\mathcal{P}(D_0, D_1, R) \\
&\stackrel{(*)}{=} \int \mathbb{E}_{\mathcal{A}}\left[\mathbb{I}\big(\mathcal{A}(D_0) \in R\big)\right] \, d\mathcal{P}(D_0, D_1, R) \\
&= \int \mathbb{P}_{\mathcal{A}}\big(\mathcal{A}(D_0) \in R\big) \, d\mathcal{P}(D_0, D_1, R),
\end{aligned}
$$

where $(*)$ followed from the independence of $\mathcal{A}$ and $\mathcal{P}$. Plugging this and the analogous expression for $\mathbb{P}(\mathcal{A}(\boldsymbol{D}_1) \in \boldsymbol{R})$ into (11) gives us that $\mathcal{A}$ is $(\varepsilon, \delta)$-LiDP. $\qquad\square$

## B.2 Auditing LiDP with Different Notions of Neighborhood

We describe how to modify the recipe of §3 for other notions of neighborhoods of datasets. The notion of the neighborhood in Definition 1 is also known as the "add-or-remove" neighborhood.

**Replace-one Neighborhood.** Two datasets $D, D' \in \mathcal{Z}^*$ are considered neighboring if $|D| = |D'|$ and $|D \setminus D'| = |D' \setminus D| = 1$. Roughly speaking, this leads to privacy guarantees that are roughly twice as strong as the add-or-remove notion of the neighborhood in Definition 1, as the worst-case sensitivity of the operation is doubled. We refer to [48, 63] for more details. Just like Definition 1, Definition 2 can also be adapted to this notion of neighborhood.

**Auditing LiDP with Replace-one Neighborhood.** The recipe of §3.2 can be slightly modified for this notion of neighborhood. The main difference is that the null hypothesis must now use $K$ canaries as well, with one fresh canary.

The alternative hypothesis is the same — we train a model on a randomized training dataset $\boldsymbol{D}_1 = D \cup \{\boldsymbol{c}_1, \ldots, \boldsymbol{c}_K\}$ augmented with $K$ random canaries drawn i.i.d. from $\mathcal{P}_{\text{canary}}$. Under the $j^{\text{th}}$ null hypothesis for each $j \in [K]$, we construct a coupled dataset $\boldsymbol{D}_{0,j} = D \cup \{\boldsymbol{c}_1, \ldots, \boldsymbol{c}_{j-1}, \boldsymbol{c}'_j, \boldsymbol{c}_{j+1}, \ldots, \boldsymbol{c}_K\}$, where $\boldsymbol{c}'_j$ is a fresh canary drawn i.i.d. from $\mathcal{P}_{\text{canary}}$. This coupling ensures that $(\boldsymbol{D}_{0,j}, \boldsymbol{D}_1)$ are neighboring with probability one. We restrict the rejection region $\boldsymbol{R}_j$ to now depend only on $\boldsymbol{c}_j$ and not in the index $j$, e.g., we test for the present of $\boldsymbol{c}_j$.[2]

Note the symmetry of the setup. Testing for the presence of $\boldsymbol{c}_j$ in $\boldsymbol{D}_{0,j}$ is exactly identical to testing for the presence of $\boldsymbol{c}'_j$ in $\boldsymbol{D}_1$. Thus, we can again rewrite the LiDP condition as

$$\frac{1}{K} \sum_{k=1}^{K} \mathbb{P}(\mathcal{A}(\boldsymbol{D}_1) \in \boldsymbol{R}_k) \quad \leq \quad \frac{e^{\varepsilon}}{m} \sum_{j=1}^{m} \mathbb{P}(\mathcal{A}(\boldsymbol{D}_1) \in \boldsymbol{R}'_j) + \delta. \qquad (12)$$

Note the subtle difference between (4) and (12): both sides depend only on $\mathcal{A}(\boldsymbol{D}_1)$ and we have completely eliminated the need to train models on $K - 1$ canaries.

From here on, the rest of the recipe is identical to §3 and Algorithm 1; we construct XBern confidence intervals for both sides of (12) and get a lower bound $\hat{\varepsilon}$.

## C Confidence Intervals for Exchangeable Bernoulli Means

We give a rigorous definition of the multivariate Exchangeable Bernoulli (XBern) distributions and derive their confidence intervals. We also give proof of the correctness of the confidence intervals.

---

[2]Note that the test could have depended on $\boldsymbol{c}'_j$ as well, but we do not need it here.

**Definition 5** (XBern Distributions). A random vector $(\boldsymbol{x}_1, \ldots, \boldsymbol{x}_K) \in \{0,1\}^K$ is said to be distributed as $\text{XBern}_K(\mu_1, \ldots, \mu_K)$ if:

- $\boldsymbol{x}_1, \ldots, \boldsymbol{x}_K$ is exchangeable, i.e., the vector $(\boldsymbol{x}_1, \ldots, \boldsymbol{x}_K)$ is identical in distribution to $(\boldsymbol{x}_{\pi(1)}, \ldots, \boldsymbol{x}_{\pi(K)})$ for any permutation $\pi : [K] \to [K]$, and,

- for each $\ell = 1, \ldots, K$, we have $\mathbb{E}[\boldsymbol{m}_\ell] = \mu_\ell$, where

$$\boldsymbol{m}_\ell := \frac{1}{\binom{K}{\ell}} \sum_{j_1 < \cdots < j_\ell \in [K]} \boldsymbol{x}_{j_1} \cdots \boldsymbol{x}_{j_\ell}. \tag{13}$$

We note that the $\text{XBern}_K$ distribution is fully determined by its $K$ moments $\mu_1, \ldots, \mu_K$. For $K = 1$, $\text{XBern}_1(\mu_1) = \text{Bernoulli}(\mu_1)$ is just the Bernoulli distribution.

The moments $\boldsymbol{m}_\ell$ satisfy a computationally efficient recurrence.

**Proposition 6.** *Let $\boldsymbol{x} \sim \text{XBern}_K(\mu_1, \ldots, \mu_K)$. We have the following for all $1 \le \ell \le K\boldsymbol{m}_1$:*

$$\boldsymbol{m}_\ell = \frac{\binom{K\boldsymbol{m}_1}{\ell}}{\binom{K}{\ell}} \qquad and \qquad \boldsymbol{m}_{\ell+1} = \boldsymbol{m}_\ell \left( \frac{K\boldsymbol{m}_1 - \ell}{K - \ell} \right). \tag{14}$$

Computing the first $\ell$ moments can thus be done in $O(K + \ell)$ time rather than $O(K^\ell)$ by naively computing the sum in (13).

*Proof of Proposition 6.* We show that this holds for any fixed vector $(x_1, \ldots, x_K) \in \{0,1\}^K$ and their corresponding moments $m_1, \ldots, m_K$ as defined in (13). Define the sums $s_1 := \sum_{j=1}^K x_j = K m_1$ and

$$s_\ell := \binom{K}{\ell} \sum_{j_1 < \cdots < j_\ell} x_{j_1} \cdots x_{j_\ell}. \tag{15}$$

We can compute $s_\ell$ by a counting argument: each term in the sum is non-zero only if all $\ell$ of the $x_{j_1}, \ldots, x_{j_\ell}$ are non-zero. The number of non-zero terms is equal to number of ways of selecting $\ell$ items from a total of $\sum_{j=1}^K x_j = s_1$ items, i.e.,

$$s_\ell = \binom{s_1}{\ell} = \binom{K m_1}{\ell}. \tag{16}$$

Combining Equations (15) and (16) completes the proof. $\square$

**Notation.** In this section, we are interested in giving confidence intervals on the mean $\mu_1$ of a $\text{XBern}_K(\mu_1, \ldots, \mu_K)$ random variable from $n$ i.i.d. observations:

$$\boldsymbol{x}^{(1)}, \ldots, \boldsymbol{x}^{(n)} \overset{\text{i.i.d.}}{\sim} \text{XBern}_K(\mu_1, \ldots, \mu_K).$$

We will define the confidence intervals using the empirical mean

$$\hat{\boldsymbol{\mu}}_1 = \frac{1}{n} \sum_{i=1}^n \boldsymbol{m}_1^{(i)} \quad \text{where} \quad \boldsymbol{m}_1^{(i)} = \frac{1}{K} \sum_{j=1}^K \boldsymbol{x}_j^{(i)},$$

as well as the higher-order moments for $\ell \in [K]$,

$$\hat{\boldsymbol{\mu}}_\ell = \frac{1}{n} \sum_{i=1}^n \boldsymbol{m}_\ell^{(i)} \quad \text{where} \quad \boldsymbol{m}_\ell^{(i)} = \frac{1}{\binom{K}{\ell}} \sum_{j_1 < \cdots < j_\ell \in [K]} \boldsymbol{x}_{j_1}^{(i)} \cdots \boldsymbol{x}_{j_\ell}^{(i)}.$$

### C.1 Non-Asymptotic Confidence Intervals

We start by giving non-asymptotic confidence intervals for the XBern distributions based on the Bernstein bound, recalled below [e.g. 20, Thm. 1.3.2].

---

**Algorithm 2** First-Order Bernstein Intervals

---

**Input:** Random vectors $\boldsymbol{x}^{(1)}, \ldots, \boldsymbol{x}^{(n)} \sim \text{XBern}_K(\mu_1, \ldots, \mu_K)$ with unknown parameters, failure probability $\beta \in (0,1)$.

**Output:** Confidence intervals $[\underline{\boldsymbol{\mu}}_1, \overline{\boldsymbol{\mu}}_1]$ such that $\mathbb{P}(\mu_1 < \underline{\boldsymbol{\mu}}_1) \leq \beta$ and $\mathbb{P}(\mu_1 > \overline{\boldsymbol{\mu}}_1) \leq \beta$.

1: Set $\underline{\boldsymbol{\mu}}_1$ as the unique solution of $x \in [0, \hat{\boldsymbol{\mu}}_1]$ such that

$$\hat{\boldsymbol{\mu}}_1 - x - \sqrt{\left(\frac{2}{n} \log \frac{1}{\beta}\right) x(1-x)} = \frac{2}{3n} \log \frac{1}{\beta}$$

   if it exists, else set $\underline{\boldsymbol{\mu}}_1 = 0$.

2: Set $\overline{\boldsymbol{\mu}}_1$ as the unique solution of $x \in [\hat{\boldsymbol{\mu}}_1, 1]$ such that

$$x - \hat{\boldsymbol{\mu}}_1 - \sqrt{\left(\frac{2}{n} \log \frac{1}{\beta}\right) x(1-x)} = \frac{2}{3n} \log \frac{1}{\beta}$$

   if it exists, else set $\overline{\boldsymbol{\mu}}_1 = 1$.

3: **return** $\underline{\boldsymbol{\mu}}_1, \overline{\boldsymbol{\mu}}_1$.

---

**Lemma 7** (Bernstein inequality). *Let $\boldsymbol{x}_1, \ldots, \boldsymbol{x}_n$ be independent random variables with $\mathbb{E}[\boldsymbol{x}_i] = 0$, $\text{Var}(\boldsymbol{x}_i) \leq \sigma^2$, and $|\boldsymbol{x}_i| \leq b$ almost surely. Let $\boldsymbol{\mu}_1 := (1/n)\sum_{i=1}^n \boldsymbol{x}_i$. For any $t > 0$, we have*

$$\mathbb{P}(\boldsymbol{\mu}_1 > t) \leq \exp\left(-\frac{nt^2}{2\sigma^2 + 2bt/3}\right).$$

*Equivalently, we have with probability at least $1 - \beta$ that*

$$\boldsymbol{\mu}_1 \leq \sqrt{\frac{2\sigma^2}{n} \log \frac{1}{\beta} + \frac{b^2}{9n^2} \log^2 \frac{1}{\beta}} + \frac{b}{3n} \log \frac{1}{\beta} \leq \sqrt{\frac{2\sigma^2}{n} \log \frac{1}{\beta}} + \frac{2b}{3n} \log \frac{1}{\beta}.$$

The last simplification is obtained by using $\sqrt{a+b} \leq \sqrt{a} + \sqrt{b}$ for scalars $a, b \geq 0$.

### C.1.1 First-Order Bernstein Intervals

The first-order Bernstein interval only depends on the empirical mean $\boldsymbol{\mu}_1$ and is given in Algorithm 2.

**Proposition 8.** *Consider Algorithm 2 with inputs $n$ i.i.d. samples $\boldsymbol{x}^{(1)}, \ldots, \boldsymbol{x}^{(n)} \overset{i.i.d.}{\sim} \text{XBern}_K(\mu_1, \ldots, \mu_K)$ for some $K \geq 1$. Then, its outputs $\underline{\boldsymbol{\mu}}_1, \overline{\boldsymbol{\mu}}_1$ satisfy $\mathbb{P}(\mu_1 \leq \underline{\boldsymbol{\mu}}_1) \geq 1 - \beta$ and $\mathbb{P}(\mu_1 \geq \overline{\boldsymbol{\mu}}_1) \geq 1 - \beta$.*

*Proof.* Applying Bernstein's inequality to $\boldsymbol{m}_1 := (1/K)\sum_{j=1}^K \boldsymbol{x}_j$, we have with probability $1 - \beta$ that

$$\mu_1 - \hat{\boldsymbol{\mu}}_1 \leq \sqrt{\frac{2\text{Var}(\boldsymbol{m}_1)}{n} \log \frac{1}{\beta}} + \frac{2}{3n} \log \frac{1}{\beta} \leq \sqrt{\frac{2\mu_1(1-\mu_1)}{n} \log \frac{1}{\beta}} + \frac{2}{3n} \log \frac{1}{\beta},$$

where we used that $\text{Var}(\boldsymbol{m}_1) \leq \mu_1(1 - \mu_1)$ since $\boldsymbol{m}_1 \in [0,1]$ a.s. We see from Figure 2 that $\overline{\boldsymbol{\mu}}_1$ is that largest value of $\mu_1$ that satisfies the above inequality, showing that it is an upper confidence bound. Similarly, we get that $\underline{\boldsymbol{\mu}}_1$ is a valid lower confidence bound. $\square$

---

**Algorithm 3** Second-Order Bernstein Intervals

---

**Input:** Random vectors $\boldsymbol{x}^{(1)}, \ldots, \boldsymbol{x}^{(n)} \sim \text{XBern}_K(\mu_1, \ldots, \mu_K)$ with unknown parameters, failure probability $\beta \in (0, 1)$.

**Output:** Confidence intervals $[\underline{\boldsymbol{\mu}}_1, \overline{\boldsymbol{\mu}}_1]$ such that $\mathbb{P}(\mu_1 < \underline{\boldsymbol{\mu}}_1) \leq \beta$ and $\mathbb{P}(\mu_1 > \overline{\boldsymbol{\mu}}_1) \leq \beta$.

1: For each $i \in [n]$, set $\boldsymbol{m}_1^{(i)} = (1/K) \sum_{j=1}^K \boldsymbol{x}_j^{(i)}$ and $\boldsymbol{m}_2^{(i)} = \boldsymbol{m}_1^{(i)} \left( \frac{K \boldsymbol{m}_1^{(i)} - 1}{K - 1} \right)$ .

2: Set $\hat{\boldsymbol{\mu}}_\ell = (1/n) \sum_{i=1}^n \boldsymbol{m}_\ell^{(i)}$ for $\ell = 1, 2$.

3: Set $\overline{\boldsymbol{\mu}}_2$ as the unique solution of $x \in [\hat{\boldsymbol{\mu}}_2, 1]$ such that

$$x - \hat{\boldsymbol{\mu}}_2 - \sqrt{\left( \frac{2}{n} \log \frac{2}{\beta} \right) x(1 - x)} = \frac{2}{3n} \log \frac{2}{\beta} \, .$$

if it exists, else set $\overline{\boldsymbol{\mu}}_2 = 1$.

4: Set $\underline{\boldsymbol{\mu}}_1$ as the unique solution of $x \in [0, \hat{\boldsymbol{\mu}}_1]$ such that

$$\hat{\boldsymbol{\mu}}_1 - x - \sqrt{\left( \frac{2}{n} \log \frac{2}{\beta} \right) \left( \frac{x}{K} - x^2 + \frac{K - 1}{K} \overline{\boldsymbol{\mu}}_2 \right)} = \frac{2}{3n} \log \frac{2}{\beta}$$

if it exists, else set $\underline{\boldsymbol{\mu}}_1 = 0$.

5: Set $\overline{\boldsymbol{\mu}}_1$ as the unique solution of $x \in [\hat{\boldsymbol{\mu}}_1, 1]$ such that

$$x - \hat{\boldsymbol{\mu}}_1 - \sqrt{\left( \frac{2}{n} \log \frac{2}{\beta} \right) \left( \frac{x}{K} - x^2 + \frac{K - 1}{K} \overline{\boldsymbol{\mu}}_2 \right)} = \frac{2}{3n} \log \frac{2}{\beta}$$

if it exists, else set $\overline{\boldsymbol{\mu}}_1 = 1$.

6: **return** $\underline{\boldsymbol{\mu}}_1, \overline{\boldsymbol{\mu}}_1$.

---

### C.1.2 Second-Order Bernstein Intervals

The second-order Bernstein interval only depends on the first two empirical moments $\boldsymbol{\mu}_1$ and $\boldsymbol{\mu}_2$. It is given in Algorithm 3. The algorithm is based on the calculation

$$
\begin{aligned}
\text{Var}(\boldsymbol{m}_1) &= \mathbb{E}[\boldsymbol{m}_1^2] - \mu_1^2 \\
&= \mathbb{E} \left[ \frac{1}{K^2} \sum_{j=1}^K \boldsymbol{x}_j^2 + \frac{2}{K^2} \sum_{j_1 < j_2 \in [K]} \boldsymbol{x}_{j_1} \boldsymbol{x}_{j_2} \right] - \mu_1^2 \\
&= \frac{\mu_1}{K} - \mu_1^2 + \frac{K - 1}{K} \mu_2 \, ,
\end{aligned}
\tag{17}
$$

where we used $\boldsymbol{x}_j^2 = \boldsymbol{x}_j$ since it is an indicator.

**Proposition 9.** *Consider Algorithm 3 with inputs $n$ i.i.d. samples $\boldsymbol{x}^{(1)}, \ldots, \boldsymbol{x}^{(n)} \overset{i.i.d.}{\sim}$ XBern$_K(\mu_1, \ldots, \mu_K)$ for some $K \geq 2$. Then, its outputs $\underline{\boldsymbol{\mu}}_1, \overline{\boldsymbol{\mu}}_1$ satisfy $\mathbb{P}(\mu_1 \geq \underline{\boldsymbol{\mu}}_1) \geq 1 - \beta$ and $\mathbb{P}(\mu_1 \leq \overline{\boldsymbol{\mu}}_1) \geq 1 - \beta$ and $\mathbb{P}(\underline{\boldsymbol{\mu}}_1 \leq \mu_1 \leq \overline{\boldsymbol{\mu}}_1) \leq 1 - \frac{3\beta}{2}$.*

*Proof.* Algorithm 3 computes the correct 2nd moment $\boldsymbol{m}_2^{(i)}$ due to Proposition 6. Applying Bernstein's inequality to $\boldsymbol{m}_2$, we get $\mathbb{P}(\mu_2 \leq \overline{\boldsymbol{\mu}}_2) \geq 1 - \beta/2$ (see also the proof of Proposition 8). Next, from Bernstein's inequality applied to $\boldsymbol{m}_1$, we leverage (17) to say that with probability at least $1 - \beta/2$, we have

$$\mu_1 - \hat{\boldsymbol{\mu}}_1 \leq \frac{2}{3n} \log \frac{2}{\beta} + \sqrt{\frac{2}{n} \log \frac{2}{\beta} \left( \frac{\mu_1}{K} - \mu_1^2 + \frac{K - 1}{K} \mu_2 \right)} \, .$$

Together with the result on $\overline{\boldsymbol{\mu}}_2$, we have with probability at least $1 - \beta$ that

$$\mu_1 - \hat{\boldsymbol{\mu}}_1 \leq \frac{2}{3n} \log \frac{2}{\beta} + \sqrt{\frac{2}{n} \log \frac{2}{\beta} \left( \frac{\mu_1}{K} - \mu_1^2 + \frac{K - 1}{K} \overline{\boldsymbol{\mu}}_2 \right)} \, .$$

---
**Algorithm 4** Fourth-Order Bernstein Intervals
---

**Input:** Random vectors $\boldsymbol{x}^{(1)}, \ldots, \boldsymbol{x}^{(n)} \sim \mathrm{XBern}_K(\mu_1, \ldots, \mu_K)$ with unknown parameters, failure probability $\beta \in (0, 1)$.

**Output:** Confidence intervals $[\underline{\boldsymbol{\mu}}_1, \overline{\boldsymbol{\mu}}_1]$ such that $\mathbb{P}(\mu_1 < \underline{\boldsymbol{\mu}}_1) \leq \beta$ and $\mathbb{P}(\mu_1 > \overline{\boldsymbol{\mu}}_1) \leq \beta$.

1: For each $i \in [n]$, set $\boldsymbol{m}_1^{(i)} = (1/K) \sum_{j=1}^K \boldsymbol{x}_j^{(i)}$ and for $\ell = 1, 2, 3$: $\boldsymbol{m}_{\ell+1}^{(i)} = \boldsymbol{m}_\ell^{(i)} \left( \frac{K \boldsymbol{m}_1^{(i)} - \ell}{K - \ell} \right)$.

2: Set $\hat{\boldsymbol{\mu}}_\ell = (1/n) \sum_{i=1}^n \boldsymbol{m}_\ell^{(i)}$ for $\ell = 1, 2, 3, 4$.

3: For $\ell = 3, 4$, set $\overline{\boldsymbol{\mu}}_\ell$ as the unique solution of $x \in [\hat{\boldsymbol{\mu}}_\ell, 1]$ such that

$$x - \hat{\boldsymbol{\mu}}_\ell - \sqrt{\left( \frac{2}{n} \log \frac{4}{\beta} \right) x(1-x)} = \frac{2}{3n} \log \frac{4}{\beta}.$$

if it exists, else set $\overline{\boldsymbol{\mu}}_\ell = 1$.

4: Set $\overline{\boldsymbol{\mu}}_2$ as the unique solution, if it exists, of $x \in [\hat{\boldsymbol{\mu}}_2, 1]$ such that

$$x - \hat{\boldsymbol{\mu}}_2 - \sqrt{\left( \frac{2}{n} \log \frac{4}{\beta} \right) \sigma_2^2 (x, \overline{\boldsymbol{\mu}}_3, \overline{\boldsymbol{\mu}}_4)} = \frac{2}{3n} \log \frac{4}{\beta}$$

where $\sigma_2^2(\cdot, \cdot, \cdot)$ is as defined in (18). Else set $\overline{\boldsymbol{\mu}}_2 = 1$.

5: Set $\underline{\boldsymbol{\mu}}_1$ as the unique solution of $x \in [0, \hat{\boldsymbol{\mu}}_1]$ such that

$$\hat{\boldsymbol{\mu}}_1 - x - \sqrt{\left( \frac{2}{n} \log \frac{4}{\beta} \right) \left( \frac{x}{k} - x^2 + \frac{K-1}{K} \overline{\boldsymbol{\mu}}_2 \right)} = \frac{2}{3n} \log \frac{4}{\beta}$$

if it exists, else set $\underline{\boldsymbol{\mu}}_1 = 0$.

6: Set $\overline{\boldsymbol{\mu}}_1$ as the unique solution of $x \in [\hat{\boldsymbol{\mu}}_1, 1]$ such that

$$x - \hat{\boldsymbol{\mu}}_1 - \sqrt{\left( \frac{2}{n} \log \frac{4}{\beta} \right) \left( \frac{x}{k} - x^2 + \frac{K-1}{K} \overline{\boldsymbol{\mu}}_2 \right)} = \frac{2}{3n} \log \frac{4}{\beta}$$

if it exists, else set $\overline{\boldsymbol{\mu}}_1 = 1$.

7: **return** $\underline{\boldsymbol{\mu}}_1, \overline{\boldsymbol{\mu}}_1$.

---

We can verify that the output $\overline{\mu}_1$ is the largest value of $\mu_1 \leq 1$ that satisfies the above inequality. Similarly, $\underline{\boldsymbol{\mu}}_1$ is obtained as a lower Bernstein bound on $\boldsymbol{m}_1$ with probability at least $1 - \beta$. By the union bound, $\underline{\boldsymbol{\mu}}_1 \leq \mu_1 \leq \overline{\boldsymbol{\mu}}_1$ holds with probability at least $1 - 3\beta/2$, since we have three invocations of Bernstein's inequality, each with a failure probability of $\beta/2$. $\qquad\square$

### C.1.3 Fourth-Order Bernstein Intervals

The fourth-order Bernstein interval depends on the first four empirical moments $\boldsymbol{\mu}_1, \ldots, \boldsymbol{\mu}_4$. It is given in Algorithm 4. The derivation of the interval is based on the calculation

$$
\begin{aligned}
\mathrm{Var}(\boldsymbol{m}_2) &= \mathbb{E}[\boldsymbol{m}_2^2] - \mu_2^2 \\
&= \frac{2\mu_2}{K(K-1)} - \mu_2^2 + \frac{4(K-2)}{K(K-1)}\mu_3 + \frac{(K-2)(K-3)}{K(K-1)}\mu_4 \\
&= \frac{2\mu_2(1-\mu_2)}{K(K-1)} + \frac{4(K-2)}{K(K-1)}(\mu_3 - \mu_2^2) + \frac{(K-2)(K-3)}{K(K-1)}(\mu_4 - \mu_2^2) \\
&=: \sigma_2^2(\mu_2, \mu_3, \mu_4).
\end{aligned}
\tag{18}
$$

**Proposition 10.** *Consider Algorithm 4 with inputs $n$ i.i.d. samples $\boldsymbol{x}^{(1)}, \ldots, \boldsymbol{x}^{(n)} \overset{i.i.d.}{\sim}$ $\mathrm{XBern}_K(\mu_1, \ldots, \mu_K)$ for some $K \geq 4$. Then, its outputs $\underline{\boldsymbol{\mu}}_1, \overline{\boldsymbol{\mu}}_1$ satisfy $\mathbb{P}(\mu_1 \geq \underline{\boldsymbol{\mu}}_1) \geq 1 - \beta$ and $\mathbb{P}(\mu_1 \leq \overline{\boldsymbol{\mu}}_1) \geq 1 - \beta$ and $\mathbb{P}(\underline{\boldsymbol{\mu}}_1 \leq \mu_1 \leq \overline{\boldsymbol{\mu}}_1) \leq 1 - \frac{5\beta}{4}$.*

---

**Algorithm 5** First-Order Wilson Intervals

---

**Input:** Random vectors $\boldsymbol{x}^{(1)}, \ldots, \boldsymbol{x}^{(n)} \sim \mathrm{XBern}_K(\mu_1, \ldots, \mu_K)$ with unknown parameters, failure probability $\beta \in (0, 1)$.

**Output:** Asymptotic confidence intervals $[\underline{\boldsymbol{\mu}}_1, \overline{\boldsymbol{\mu}}_1]$ such that $\lim_{n \to \infty} \mathbb{P}(\mu_1 < \underline{\boldsymbol{\mu}}_1) \leq \beta$ and $\lim_{n \to \infty} \mathbb{P}(\mu_1 > \overline{\boldsymbol{\mu}}_1) \leq \beta$.

1: Set $\underline{\boldsymbol{\mu}}_1 < \overline{\boldsymbol{\mu}}_1$ as the roots of the quadratic in $x$:

$$(n + Z_\beta^2)\, x^2 - (2n\hat{\boldsymbol{\mu}}_1 + Z_\beta^2)\, x + n\hat{\boldsymbol{\mu}}_1^2 = 0 \,.$$

2: **return** $\underline{\boldsymbol{\mu}}_1, \overline{\boldsymbol{\mu}}_1$.

---

---

**Algorithm 6** Second-Order Wilson Intervals

---

**Input:** Random vectors $\boldsymbol{x}^{(1)}, \ldots, \boldsymbol{x}^{(n)} \sim \mathrm{XBern}_K(\mu_1, \ldots, \mu_K)$ with unknown parameters, failure probability $\beta \in (0, 1)$.

**Output:** Asymptotic confidence intervals $[\underline{\boldsymbol{\mu}}_1, \overline{\boldsymbol{\mu}}_1]$ such that $\lim_{n \to \infty} \mathbb{P}(\mu_1 < \underline{\boldsymbol{\mu}}_1) \leq \beta$ and $\lim_{n \to \infty} \mathbb{P}(\mu_1 > \overline{\boldsymbol{\mu}}_1) \leq \beta$.

1: For each $i \in [n]$, set $\boldsymbol{m}_1^{(i)} = (1/K) \sum_{j=1}^K \boldsymbol{x}_j^{(i)}$ and $\boldsymbol{m}_2^{(i)} = \boldsymbol{m}_1^{(i)} \left( \frac{K\boldsymbol{m}_1^{(i)} - 1}{K - 1} \right)$.

2: Set $\hat{\boldsymbol{\mu}}_\ell = (1/n) \sum_{i=1}^n \boldsymbol{m}_\ell^{(i)}$ for $\ell = 1, 2$.

3: Let $\overline{\boldsymbol{\mu}}_2$ be the larger root of the quadratic in $x$:

$$(n + Z_{\beta/2}^2)\, x^2 - (2n\hat{\boldsymbol{\mu}}_2 + Z_{\beta/2}^2)\, x + n\hat{\boldsymbol{\mu}}_2^2 = 0 \,.$$

4: Set $\underline{\boldsymbol{\mu}}_1 < \overline{\boldsymbol{\mu}}_1$ as the roots of the quadratic in $x$:

$$(n + Z_{\beta/2}^2)\, x^2 - \left( 2n\hat{\boldsymbol{\mu}}_1 + \frac{Z_{\beta/2}^2}{K} \right) x + n\hat{\boldsymbol{\mu}}_1^2 - \left( \frac{K-1}{K} \right) Z_{\beta/2}^2\, \overline{\boldsymbol{\mu}}_2 = 0 \,.$$

5: **return** $\underline{\boldsymbol{\mu}}_1, \overline{\boldsymbol{\mu}}_1$.

---

*Proof.* Algorithm 4 computes the correct moments $\boldsymbol{m}_\ell^{(i)}$ for $\ell \leq 4$ due to Proposition 6. Applying Bernstein's inequality to $\boldsymbol{m}_3$ and $\boldsymbol{m}_4$, we get $\mathbb{P}(\mu_\ell \leq \overline{\boldsymbol{\mu}}_\ell) \geq 1 - \beta/4$ for $\ell = 3, 4$ (see also the proof of Proposition 8).

Next, from Bernstein's inequality applied to $\boldsymbol{m}_2$, we have with probability at least $1 - \beta/4$ that

$$\mu_2 - \hat{\boldsymbol{\mu}}_2 \leq \frac{2}{3n} \log \frac{4}{\beta} + \sqrt{\frac{2\sigma_2^2(\mu_2, \mu_3, \mu_4)}{n} \log \frac{4}{\beta}} \,.$$

Combining this with the results on $\overline{\boldsymbol{\mu}}_3, \overline{\boldsymbol{\mu}}_4$ with the union bound, we get with probability at least $1 - 3\beta/4$ that

$$\mu_2 - \hat{\boldsymbol{\mu}}_2 \leq \frac{2}{3n} \log \frac{4}{\beta} + \sqrt{\frac{2\sigma_2^2(\mu_2, \overline{\boldsymbol{\mu}}_3, \overline{\boldsymbol{\mu}}_4)}{n} \log \frac{4}{\beta}} \,.$$

Finally, plugging this into a Bernstein bound on $\boldsymbol{m}_1$ using the variance calculation from (17) (also see the proof of Proposition 9) completes the proof. $\square$

## C.2  Asymptotic Confidence Intervals

We derive asymptotic versions of the Algorithms 2 to 4 using the Wilson confidence interval.

The Wilson confidence interval is a tightening of the constants for the Bernstein confidence interval

$$\mu_1 - \hat{\boldsymbol{\mu}}_1 \leq \sqrt{\frac{2\log(1/\beta)}{n} \mathrm{Var}(\boldsymbol{m}_1)} + \frac{2}{3n} \log \frac{1}{\beta} \quad \text{to} \quad \mu_1 - \hat{\boldsymbol{\mu}}_1 \leq \sqrt{\frac{Z_\beta^2}{n} \mathrm{Var}(\boldsymbol{m}_1)} \,,$$

where $Z_\beta$ is the $(1 - \beta)$-quantile of the standard Gaussian. Essentially, this completely eliminates the $1/n$ term, while the coefficient of the $1/\sqrt{n}$ term improves from $\sqrt{2\log(1/\beta)}$ to $Z_\beta$ — see Figure 2.

---

**Algorithm 7** Fourth-Order Wilson Intervals

---

**Input:** Random vectors $\boldsymbol{x}^{(1)}, \ldots, \boldsymbol{x}^{(n)} \sim \mathrm{XBern}_K(\mu_1, \ldots, \mu_K)$ with unknown parameters, failure probability $\beta \in (0, 1)$.

**Output:** Asymptotic confidence intervals $[\underline{\boldsymbol{\mu}}_1, \overline{\boldsymbol{\mu}}_1]$ such that $\lim_{n \to \infty} \mathbb{P}(\mu_1 < \underline{\boldsymbol{\mu}}_1) \leq \beta$ and $\lim_{n \to \infty} \mathbb{P}(\mu_1 > \overline{\boldsymbol{\mu}}_1) \leq \beta$.

1: For each $i \in [n]$, set $\boldsymbol{m}_1^{(i)} = (1/K) \sum_{j=1}^{K} \boldsymbol{x}_j^{(i)}$ and for $\ell = 1, 2, 3$: $\boldsymbol{m}_{\ell+1}^{(i)} = \boldsymbol{m}_\ell^{(i)} \left( \frac{K \boldsymbol{m}_1^{(i)} - \ell}{K - \ell} \right)$.

2: Set $\hat{\boldsymbol{\mu}}_\ell = (1/n) \sum_{i=1}^{n} \boldsymbol{m}_\ell^{(i)}$ for $\ell = 1, 2, 3, 4$.

3: For $\ell = 3, 4$, let $\overline{\boldsymbol{\mu}}_\ell$ be the larger root of the quadratic in $x$:

$$(n + Z_{\beta/4}^2) x^2 - (2n\hat{\boldsymbol{\mu}}_\ell + Z_{\beta/4}^2) x + n\hat{\boldsymbol{\mu}}_\ell^2 = 0.$$

4: Let $\overline{\boldsymbol{\mu}}_2$ be the larger root of the quadratic in $x$:

$$\left( n + \frac{2Z_{\beta/4}^2(2K-3)}{K(K-1)} \right) x^2 - \left( 2n\hat{\boldsymbol{\mu}}_2 + \frac{2Z_{\beta/4}^2}{K(K-1)} \right) x + n\hat{\boldsymbol{\mu}}_2^2 - cZ_{\beta/4}^2 = 0,$$

$$\text{where} \quad c = \frac{(K-2)(K-3)}{K(K-1)}(\overline{\boldsymbol{\mu}}_4 - \overline{\boldsymbol{\mu}}_3^2) + \frac{4(K-2)}{K(K-1)}\overline{\boldsymbol{\mu}}_3$$

5: Set $\underline{\boldsymbol{\mu}}_1 < \overline{\boldsymbol{\mu}}_1$ as the roots of the quadratic in $x$:

$$(n + Z_{\beta/4}^2) x^2 - \left( 2n\hat{\boldsymbol{\mu}}_1 + \frac{Z_{\beta/4}^2}{K} \right) x + n\hat{\boldsymbol{\mu}}_1^2 - \left( \frac{K-1}{K} \right) Z_{\beta/4}^2 \overline{\boldsymbol{\mu}}_2 = 0.$$

6: **return** $\underline{\boldsymbol{\mu}}_1, \overline{\boldsymbol{\mu}}_1$.

---

The Wilson approximation holds under the assumption that $(\mu_1 - \hat{\boldsymbol{\mu}}_1)/\sqrt{\mathrm{Var}(\boldsymbol{m}_1)/n} \stackrel{\mathrm{d}}{\approx} \mathcal{N}(0, 1)$ and using a Gaussian confidence interval. This can be formalized by the central limit theorem.

**Lemma 11** (Lindeberg–Lévy Central Limit Theorem). *Consider a sequence of independent random variables $\boldsymbol{y}^{(1)}, \boldsymbol{y}^{(2)}, \ldots$ with finite moments $\mathbb{E}[\boldsymbol{y}^{(i)}] = \mu < \infty$ and $\mathbb{E}(\boldsymbol{y}^{(i)} - \mu)^2 = \sigma^2 < \infty$ for each $i$. Then, the empirical mean $\hat{\boldsymbol{\mu}}_1^{(n)} = (1/n) \sum_{i=1}^{n} \boldsymbol{y}^{(i)}$ based on $n$ samples satisfies*

$$\lim_{n \to \infty} \mathbb{P} \left( \frac{\hat{\boldsymbol{\mu}}_1^{(n)} - \mu}{\sigma/\sqrt{n}} > t \right) = \mathbb{P}_{\boldsymbol{\xi} \sim \mathcal{N}(0,1)} (\boldsymbol{\xi} > t)$$

*for all $t \in \mathbb{R}$. Consequently, we have,*

$$\lim_{n \to \infty} \mathbb{P} \left( \mu - \hat{\boldsymbol{\mu}}_1^{(n)} > \sigma Z_\beta/\sqrt{n} \right) \geq 1 - \beta \quad \text{and} \quad \lim_{n \to \infty} \mathbb{P} \left( \hat{\boldsymbol{\mu}}_1^{(n)} - \mu > \sigma Z_\beta/\sqrt{n} \right) \geq 1 - \beta. \tag{19}$$

The finite moment requirement above is satisfied in our case because all our random variables are bounded between 0 and 1. The asymptotic confidence interval implied by (19) is known as the **Wilson confidence interval**. It states that

$$|\mu - \hat{\boldsymbol{\mu}}_1^{(n)}| \leq \frac{\sigma Z_{\beta/2}}{\sqrt{n}}.$$

w.p. $1 - \beta$ as $n \to \infty$.

We give the Wilson-variants of Algorithms 2 to 4 respectively in Algorithms 5 to 7. Apart from the fact that the Wilson intervals are tighter, we can also solve the equations associated with the Wilson intervals in closed form as they are simply quadratic equations (i.e., without the need for numerical root-finding). The following proposition shows their correctness.

**Proposition 12.** *Consider $n$ i.i.d. samples $\boldsymbol{x}^{(1)}, \ldots, \boldsymbol{x}^{(n)} \stackrel{i.i.d.}{\sim} \mathrm{XBern}_K(\mu_1, \ldots, \mu_K)$ as inputs to Algorithms 5 to 7. Then, their outputs $\underline{\boldsymbol{\mu}}_1, \overline{\boldsymbol{\mu}}_1$ satisfy $\lim_{n \to \infty} \mathbb{P}(\mu_1 \leq \underline{\boldsymbol{\mu}}_1) \geq 1 - \beta$ and $\lim_{n \to \infty} \mathbb{P}(\mu_1 \geq \overline{\boldsymbol{\mu}}_1) \geq 1 - \beta$ and $\lim_{n \to \infty} \mathbb{P}(\underline{\boldsymbol{\mu}}_1 \leq \mu_1 \leq \overline{\boldsymbol{\mu}}_1) \leq 1 - C\beta$ if*

(a) $K \geq 1$ and $C = 2$ for Algorithm 5,

(b) $K \geq 2$ and $C = 3/2$ for Algorithm 6, and

(c) $K \geq 4$ and $C = 5/4$ for Algorithm 7.

We omit the proof as it is identical to those of Propositions 8 to 10 except that it uses the Wilson interval from Lemma 11 rather than the Bernstein interval.

### C.3 Scaling of Higher-Order Bernstein Bounds (Proposition 4)

We now re-state and prove Proposition 4.

**Proposition 4.** *For any positive integer $\ell$ that is a power of two and $K = \lceil n^{(\ell-1)/\ell} \rceil$, suppose we have $n$ samples from a $K$-dimensional XBern distribution with parameters $(\mu_1, \ldots, \mu_K)$. If all $\ell'^{th}$-order correlations scale as $1/K$, i.e., $|\mu_{2\ell'} - \mu_{\ell'}^2| = O(1/K)$, for all $\ell' \leq \ell$ and $\ell'$ is a power of two, then the $\ell^{th}$-order Bernstein bound is $|\mu_1 - \hat{\mu}_1| = O(1/n^{(2\ell-1)/(2\ell)})$.*

*Proof.* We are given $n$ samples from an XBern distribution $\boldsymbol{x} \in \{0, 1\}^K$ with parameters $(\mu_1, \ldots, \mu_K)$, where $\mu_\ell := \mathbb{E}[\boldsymbol{m}_\ell]$ with

$$\boldsymbol{m}_\ell := \frac{1}{K(K-1)\cdots(K-\ell+1)} \sum_{j_1 < j_2 < \ldots < j_\ell \in [K]} \boldsymbol{x}_{j_1} \cdots \boldsymbol{x}_{j_\ell}.$$

By exchangeability, it also holds that $\mu_\ell = \mathbb{E}[\boldsymbol{x}_1 \cdots \boldsymbol{x}_\ell]$. Assuming a confidence level $1 - \beta < 1$ and $K = \lceil n^{(\ell-1)/\ell} \rceil$, the 1st-order Bernstein bound gives

$$|\mu_1 - \hat{\boldsymbol{\mu}}_1| = O\left(\sqrt{\frac{\sigma_1^2}{n}}\right), \tag{20}$$

where $\sigma_\ell^2 := \mathrm{Var}(\boldsymbol{m}_\ell)$. Expanding $\sigma_\ell^2$, it is easy to show that it is dominated by the $2\ell$th-order correlation $|\mu_{2\ell} - \mu_\ell^2|$:

$$\sigma_\ell^2 = O\left(\frac{1}{K} + |\mu_{2\ell} - \mu_\ell^2|\right) = O\left(\frac{1}{K} + |\hat{\boldsymbol{\mu}}_{2\ell} - \hat{\boldsymbol{\mu}}_\ell^2| + \sqrt{\frac{\sigma_{2\ell}^2}{n}}\right).$$

Note that our Bernstein confidence interval does not use the fact that the higher-order correlations are small. We only use that assumption to bound the resulting size of the confidence interval in the analysis. Applying the assumption that all the higher order correlations are bounded by $1/K$, i.e., $|\mu_{2\ell'} - \mu_{\ell'}^2| = O(1/K)$, we get that $|\hat{\boldsymbol{\mu}}_{2\ell'} - \hat{\boldsymbol{\mu}}_{\ell'}^2| = O(1/K + \sqrt{\sigma_{2\ell'}^2/n})$. Applying this recursively into (20), we get that

$$|\mu_1 - \hat{\boldsymbol{\mu}}_1| = O\left(\sqrt{\frac{1}{nK}} + \frac{\sigma_\ell^{1/\ell}}{n^{(2\ell-1)/(2\ell)}}\right),$$

for any $\ell$ that is a power of two. For an $\ell^{\text{th}}$-order Bernstein bound, we only use moment estimates up to $\ell$ and bound $\sigma_\ell^2 \leq 1$. The choice of $K = n^{(\ell-1)/\ell}$ gives the desired bound: $|\mu_1 - \hat{\boldsymbol{\mu}}_1| = O(1/n^{(2\ell-1)/(2\ell)})$. $\qquad\square$

## D Canary Design for Lifted DP: Details

The canary design employed in the auditing of the usual $(\varepsilon, \delta)$-DP can be easily extended to create distributions over canaries to audit LiDP. We give some examples for common classes of canaries.

**Setup.** We assume a supervised learning setting with a training dataset $D_{\text{train}} = \{(x_i, y_i)\}_{i=1}^N$ and a held-out dataset $D_{\text{val}} = \{(x_i, y_i)\}_{i=N+1}^{N+N'}$ of pairs of input $x_i \in \mathcal{X}$ and output $y_i \in \mathcal{Y}$. We then aim to minimize the average loss

$$F(\theta) = \frac{1}{N} \sum_{i=1}^N L((x_i, y_i), \theta), \tag{21}$$

where $L(z, \theta)$ is the loss incurred by model $\theta$ on input-output pair $z = (x, y)$.

In the presence of canaries $c_1, \ldots, c_k$, we instead aim to minimize the objective

$$F_{\text{canary}}(\theta; c_1, \ldots, c_k) = \frac{1}{N} \left( \sum_{i=1}^{N} L((x_i, y_i), \theta) + \sum_{j=1}^{k} L_{\text{canary}}(c_j, \theta) \right), \qquad (22)$$

where $L_{\text{canary}}(c, \theta)$ is the loss function for a canary $c$ — this may or may not coincide with the usual loss $L$.

**Goals of Auditing DP.** The usual practice is to set $D_0 = D_{\text{train}}$ and $D_1 = D_0 \cup \{c\}$ and $R \equiv R_c$ for a canary $c$. Recall from the definition of $(\varepsilon, \delta)$-DP in (1), we have

$$\varepsilon \geq \sup_{c \in \mathcal{C}} \log \left( \frac{\mathbb{P}(\mathcal{A}(D_1) \in R_c) - \delta}{\mathbb{P}(\mathcal{A}(D_0) \in R_c)} \right),$$

for some class $\mathcal{C}$ of canaries. The goal then is to find canaries $c$ that approximate the sup over $\mathcal{C}$. Since this goal is hard, one usually resorts to finding canaries whose effect can be "easy to detect" in some sense.

**Goals of Auditing LiDP.** Let $P_{\text{canary}}$ denote a probability distribution over a set $\mathcal{C}$ of allowed canaries. We sample $K$ canaries $\boldsymbol{c}_1, \ldots, \boldsymbol{c}_K \overset{\text{i.i.d.}}{\sim} P_{\text{canary}}$ and set

$$\boldsymbol{D}_0 = D_{\text{train}} \cup \{\boldsymbol{c}_1, \ldots, \boldsymbol{c}_{K-1}\}, \quad \boldsymbol{D}_1 = D_{\text{train}} \cup \{\boldsymbol{c}_1, \ldots, \boldsymbol{c}_K\}, \quad \boldsymbol{R} = R_{\boldsymbol{c}_K}.$$

From the definition of $(\varepsilon, \delta)$-LiDP in (3), we have

$$\varepsilon \geq \sup_{P_{\text{canary}}} \log \left( \frac{\mathbb{P}(\mathcal{A}(\boldsymbol{D}_1) \in \boldsymbol{R}) - \delta}{\mathbb{P}(\mathcal{A}(\boldsymbol{D}_0) \in \boldsymbol{R})} \right),$$

for some class $\mathcal{C}$ of canaries. The goal then is to approximate the distribution $P_{\text{canary}}$ for each choice of the canary set $\mathcal{C}$. Since this is hard, we will attempt to define a distribution over canaries that are easy to detect (similar to the case of auditing DP). Following the discussion in §3.3, auditing LiDP benefits the most when the canaries are uncorrelated. To this end, we will also impose the restriction that a canary $\boldsymbol{c} \sim P_{\text{canary}}$, if included in the training of a model $\theta$, is unlikely to change the membership of $\theta \in R_{\boldsymbol{c}'}$ for an i.i.d. canary $\boldsymbol{c}' \sim P_{\text{canary}}$ that is independent of $\boldsymbol{c}$.

We consider two choices of the canary set $\mathcal{C}$ (as well as the outcome set $R_c$ and the loss $L_{\text{canary}}$): data poisoning, and random gradients.

### D.1 Data Poisoning

We describe the data poisoning approach known as ClipBKD [32] that is based on using the tail singular vectors of the input data matrix and its extension to auditing LiDP.

Let $X = (x_1^\top; \cdots; x_N^\top) \in \mathbb{R}^{N \times d}$ denote the matrix with the datapoints $x_i \in \mathbb{R}^d$ as rows. Let $X = \sum_{i=1}^{\min\{N, d\}} \sigma_i u_i v_i^\top$ be the singular value decomposition of $X$ with $\sigma_1 \leq \sigma_2 \leq \cdots$ be the singular values arranged in ascending order. Let $Y$ denote the set of allowed labels.

For this section, we take the set of allowed canaries $\mathcal{C} = \{\alpha v_1, \alpha v_2, \ldots, \alpha v_{\min\{N, d\}}\} \times \mathcal{Y}$ as the set of the right singular vector of $X$ scaled by a given factor $\alpha > 0$ together with any possible target from $\mathcal{Y}$. We take $L_{\text{canary}}(c, \theta) = L(c, \theta)$ to be the usual loss function, and the output set $R_c$ to be the loss-thresholded set

$$R_c := \{\theta \in \mathcal{R} \,:\, L(c, \theta) \leq \tau\}, \qquad (23)$$

for some threshold $\tau$.

**Auditing DP.** The ClipBKD approach [32] uses a canary with input $\alpha v_1$, the singular vector corresponding to the smallest singular value, scaled by a parameter $\alpha > 0$. The label is taken as $y^\star(\alpha v_1)$, where

$$y^\star(x) = \arg\max_{y \in \mathcal{Y}} L((x, y), \theta_0^\star)$$

is the target that has the highest loss on input $x$ under the empirical risk minimizer $\theta_0^\star = \arg\min_{\theta \in \mathcal{R}} F(\theta)$. Since a unique $\theta_0^\star$ is not guaranteed for deep nets nor can we find it exactly,

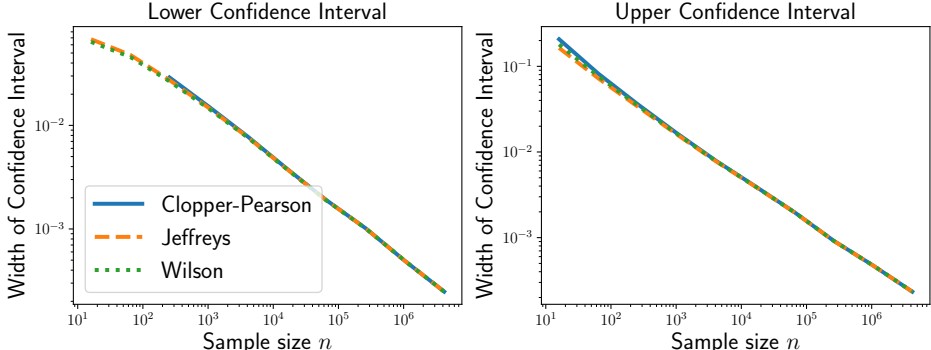

Figure 7: Comparing the Binomial proportion confidence intervals. We sample $m \sim \mathrm{Binomial}(n, p)$ for $p = 0.1$ and $n$ varying and find the $95\%$ confidence interval $[\underline{\boldsymbol{p}}_n, \overline{\boldsymbol{p}}_n]$. We plot the widths $p - \underline{\boldsymbol{p}}_n$ and $\overline{\boldsymbol{p}}_n - p$ versus $n$. We find that all confidence intervals are nearly equivalent once $n$ is larger than $\approx 1/\min\{p, 1-p\}^2$.

we train 100 models with different random seeds and pick the class $y$ that yields the highest average loss over these runs.

**Auditing LiDP.** We extend ClipBKD to define a probability distribution over a given number $p$ of canaries. We take

$$P_{\mathrm{canary}} = \mathrm{Uniform}\big(\{c_1, \ldots, c_p\}\big) \quad \text{with} \quad c_j = \big(\alpha v_j, y^\star(\alpha v_j)\big), \tag{24}$$

i.e., $P_{\mathrm{canary}}$ is the uniform distribution over the $p$ singular vectors corresponding to the smallest singular values.

### D.2 Random Gradients

The update poisoning approach of [2] relies of supplying gradients $c \sim \mathrm{Uniform}(B_{\mathcal{R}}(0, r))$ that are uniform on the Euclidean ball of a given radius $r$. This is achieved by setting the loss of the canary as

$$L_{\mathrm{canary}}(c, \theta) = \langle c, \theta \rangle, \quad \text{so that} \quad \nabla_\theta L_{\mathrm{canary}}(c, \theta) = c,$$

is the desired vector $c$.

The set $R_c$ is a threshold of the dot product

$$R_c = \{\theta \in \mathcal{R} \ : \ \langle c, \theta \rangle \leq \tau\} \tag{25}$$

for a given threshold $\tau$. This set is analogous to the loss-based thresholding of (23) in that both can be written as $L_{\mathrm{canary}}(c, \theta) \leq \tau$.

**Auditing DP and LiDP.** The random gradient approach of [2] relies on defining a distribution $P_{\mathrm{canary}} \equiv \mathrm{Uniform}(B_{\mathcal{R}}(0, r))$ over canaries. It can be used directly to audit LiDP.

## E Simulations with the Gaussian Mechanism: Details and More Results

Here, we give the full details and additional results of auditing the Gaussian mechanism with synthetic data in §4.

### E.1 Experiment Setup

Fix a dimension $d$ and a failure probability $\beta \in (0, 1)$. Suppose we have a randomized algorithm $\mathcal{A}$ that returns a noisy sum of its inputs with the goal of simulating the Gaussian mechanism. Concretely, the input space $\mathcal{Z} = \{z \in \mathbb{R}^d \ : \ \|z\|_2 \leq 1\}$ is the unit ball in $\mathbb{R}^d$. Given a finite set $D \in \mathcal{Z}^*$, we sample a vector $\boldsymbol{\xi} \sim \mathcal{N}(0, \sigma^2 \boldsymbol{I}_d)$ of a given variance $\sigma^2$ and return

$$\mathcal{A}(D) = \boldsymbol{\xi} + \sum_{z \in D} z \,.$$

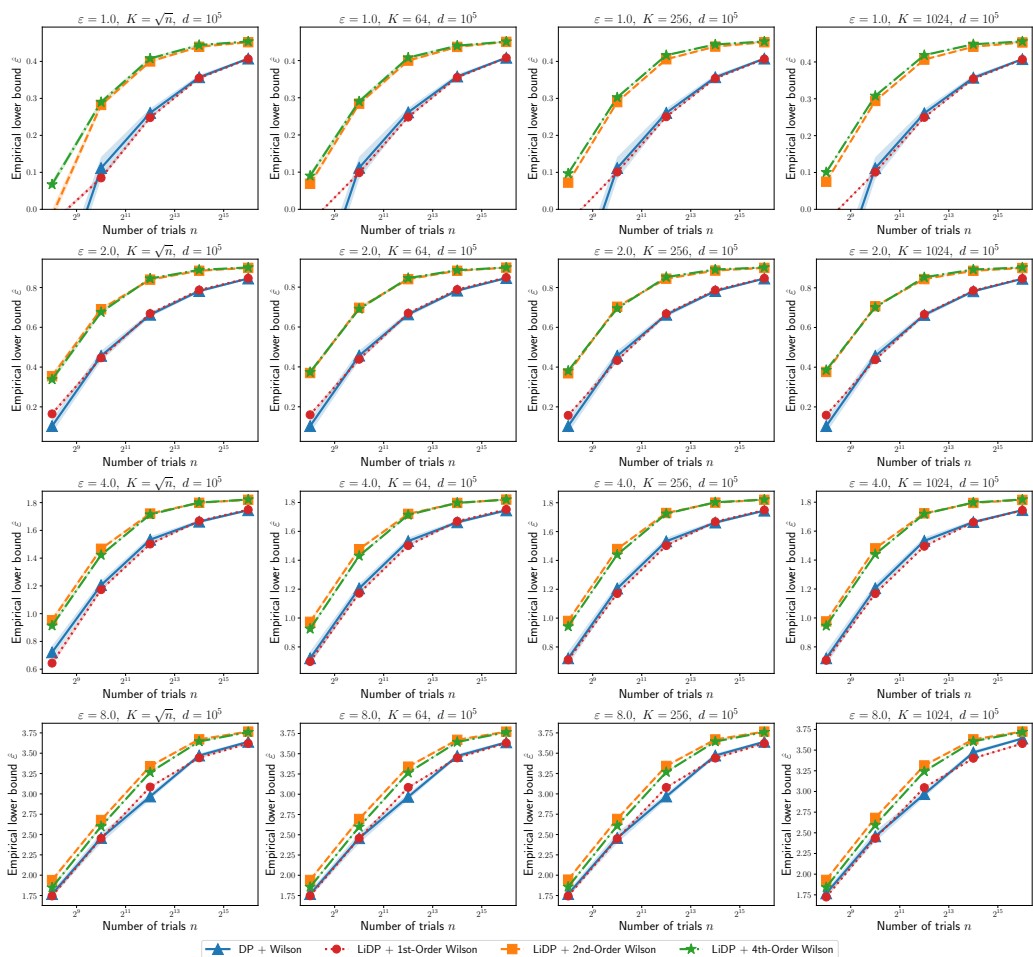

Figure 8: Effect of the number $n$ of trials on the empirical lower bound $\hat{\varepsilon}$ from auditing the Gaussian mechanism for DP and LiDP. The shaded are denotes the standard error over 25 random seeds.

To isolate the effect of the canaries, we set our original dataset $D = \{\mathbf{0}_d\}$ as a singleton with the vector of zeros in $\mathbb{R}^d$. Since we are in the blackbox setting, we do not assume that this is known to the auditor.

**DP Upper Bound.** The non-private version of our function computes the sum $D \mapsto \sum_{z \in D} z$. where each $z \in D$ is a canary. Hence, the $\ell_2$ sensitivity of the operation is $\Delta_2 = \max_{x \in D} \|x\|_2 = 1$, as stated in §4.

Since we add $\boldsymbol{\xi} \sim \mathcal{N}(0, \sigma^2 I_d)$, it follows that the operation $\mathcal{A}(\cdot)$ is $\left(\alpha, \alpha/(2\sigma^2)\right)$-RDP for every $\alpha > 1$. Thus, $\mathcal{A}(\cdot)$ is $(\varepsilon_\delta, \delta)$-DP where

$$\varepsilon_\delta \leq \inf_{\alpha > 1} \left\{ \frac{\alpha}{2\sigma^2} + \frac{1}{\alpha - 1} \log \frac{1}{\alpha\delta} + \log \left( 1 - \frac{1}{\alpha} \right) \right\} ,$$

based on [4, Thm. 21]. This can be shown to be bounded above by $\frac{1}{\sigma}\sqrt{2 \log \frac{1}{\delta}} + \frac{1}{2\sigma^2}$ [44, Prop. 3]. By Theorem 3, it follows that the operation $\mathcal{A}(\cdot)$ is also $(\varepsilon_\delta, \delta)$-LiDP.

**Auditing LiDP.** We follow the recipe of Algorithm 1. We set the rejection region $\boldsymbol{R} = R_\tau(\boldsymbol{c}_K)$ as a function of the canary $\boldsymbol{c}_K$ that differs between $\boldsymbol{D}_0$ and $\boldsymbol{D}_1$, where

$$R_\tau(\boldsymbol{c}_j) := \left\{ u \in \mathbb{R}^d \ : \ \langle u, \boldsymbol{c}_j \rangle \geq \tau \right\} , \tag{26}$$

and $\tau \in \mathbb{R}$ is a tuned threshold.

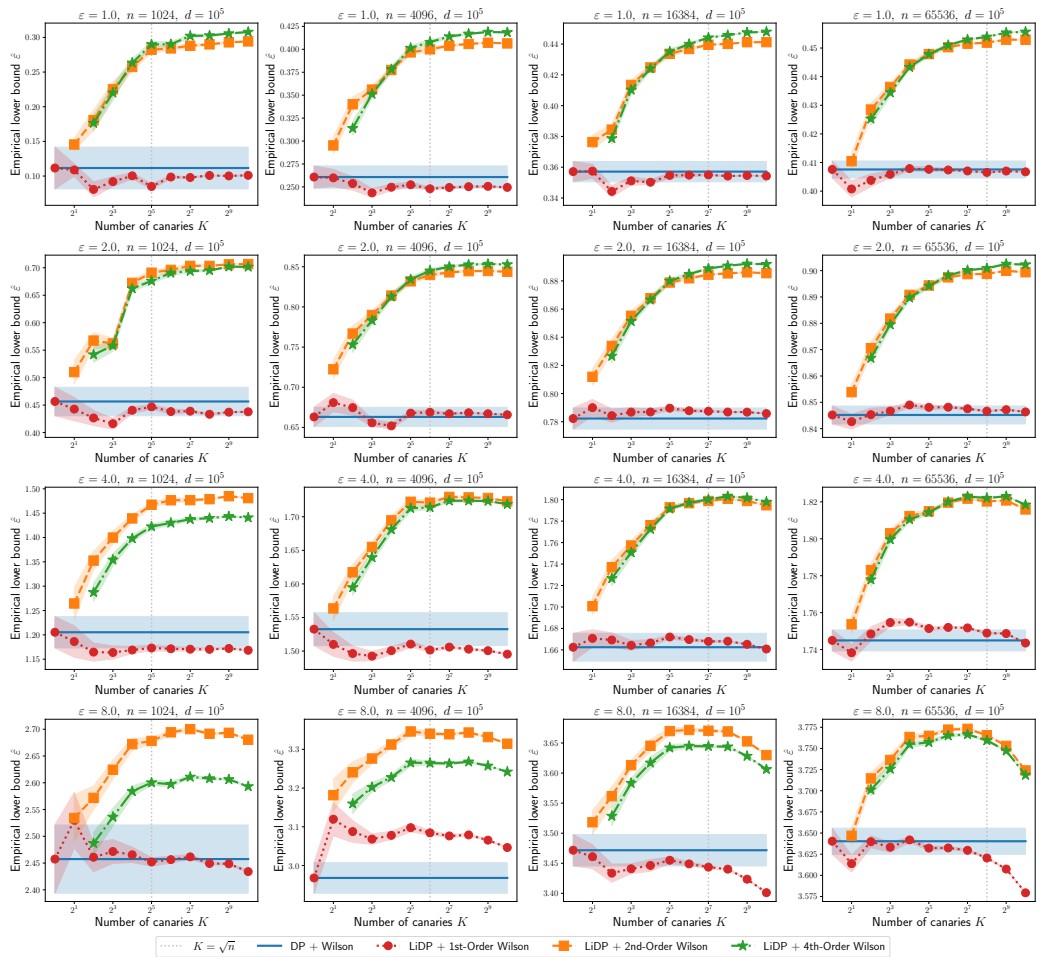

Figure 9: Effect of the number $k$ of canaries on the empirical lower bound $\hat{\varepsilon}$ from auditing the Gaussian mechanism for DP and LiDP. The shaded are denotes the standard error over 25 random seeds.

We evaluate empirical privacy auditing methods by how large the lower bound $\hat{\varepsilon}$ is — the higher the lower bound, the better the confidence interval.

**Methods Compared.** An empirical privacy auditing method is defined by the type of privacy auditing (DP or LiDP) and the type of confidence intervals. We compare the following auditing methods:

- **DP + Wilson**: We audit the usual $(\varepsilon, \delta)$-DP with $K = 1$ canary. This corresponds exactly to auditing LiDP with $K = 1$. We use the 1st-Order Wilson confidence intervals for a fair comparison with the other LiDP auditing methods. This performs quite similarly to the other intervals used in the literature, cf. Figure 7.

- **LiDP + 1st-Order Wilson**: We audit LiDP with $K$ canaries with the 1st-Order Wilson confidence interval. This method cannot leverage the shrinking of the confidence intervals from higher-order estimates.

- **LiDP + 2nd/4th-Order Wilson**: We audit LiDP with $K > 1$ canaries using the higher-order Wilson confidence intervals. Unless stated otherwise, we use $m = K$ test canaries.

**Parameters of the Experiment.** We vary the following parameters in the experiment:

- Number of trials $n \in \{2^8, 2^{10}, \cdots, 2^{16}\}$.
- Number of canaries $k \in \{1, 2, 2^2, \ldots, 2^{10}\}$.
- Dimension $d \in \{10^2, 10^3, \ldots, 10^6\}$.

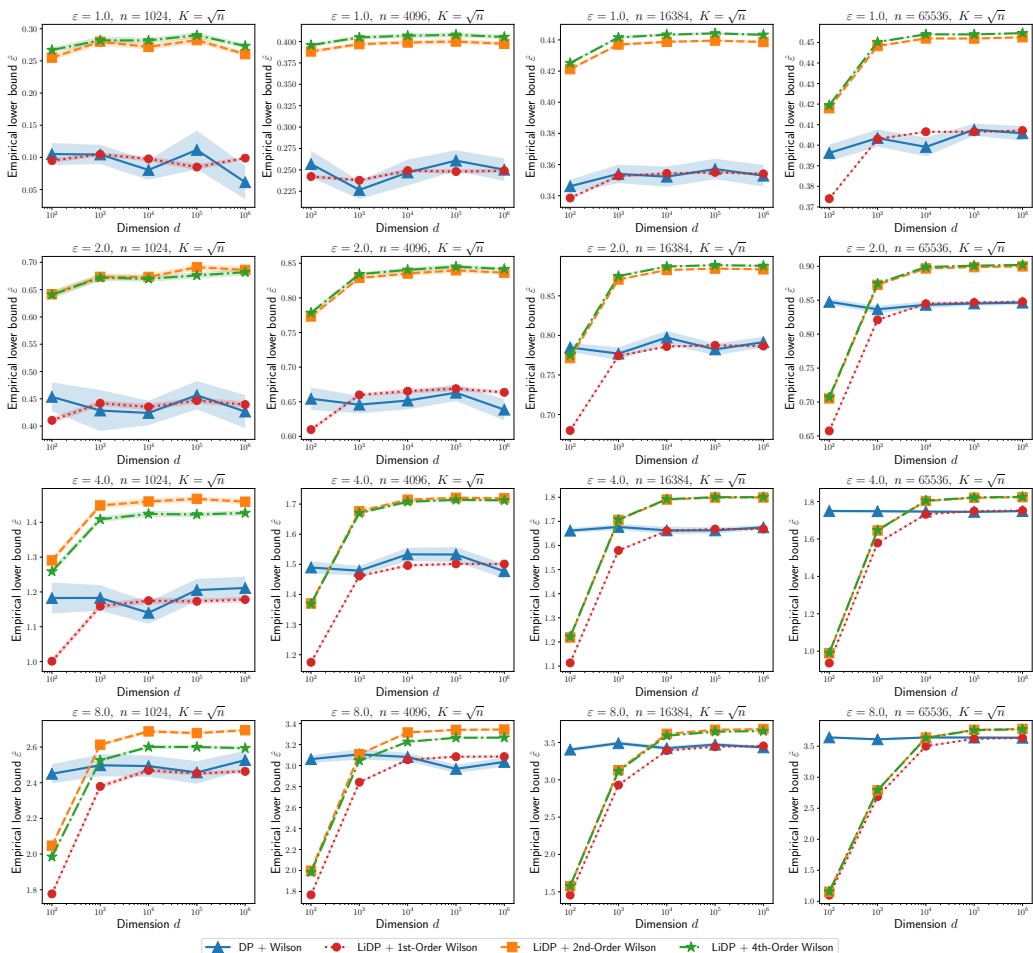

Figure 10: Effect of the data dimension $d$ on the empirical lower bound $\hat\varepsilon$ from auditing the Gaussian mechanism for DP and LiDP. The shaded area denotes the standard error over 25 random seeds.

- DP upper bound $\varepsilon \in \{1, 2, 4, 8\}$.

We fix the DP parameter $\delta = 10^{-5}$ and the failure probability $\beta = 0.05$.

**Tuning the threshold $\tau$.** For each confidence interval scheme, we repeat the estimation of the lower bound $\hat\varepsilon(\tau)$ for a grid of thresholds $\tau \in \Gamma$ on a holdout set of $n$ trials. We fix the best threshold $\tau^* = \arg\max_{\tau \in \Gamma} \hat\varepsilon(\tau)$ that gives the largest lower bound $\hat\varepsilon(\tau)$ from the grid $\Gamma$. We then fix the threshold $\tau^*$ and report numbers over a fresh set of $n$ trials.

**Randomness and Repetitions.** We repeat each experiment 25 times (after fixing the threshold) with different random seeds and report the mean and standard error.

## E.2 Additional Experimental Results

**Effect of the Number $m$ of Test Canaries.** In Figure 13, we vary the number $m$ of test canaries while keeping the number $K$ of training canaries fixed. We see that the 2nd and 4th-order Wilson estimators lead to monotonically improving lower bound $\hat\varepsilon$ as $m$ increases. However, these improvements soon flatten out between $m = 8$ and $m = 32$ depending on other parameters. In particular, the default value of $m = K$ is nearly or equal to the best value of $m$ for $K \geq 16$.

We give additional experimental results, expanding on the plots shown in Figures 3 and 4:

- Figure 8 shows the effect of varying the number of trials $n$, similar to Figure 3 (left).

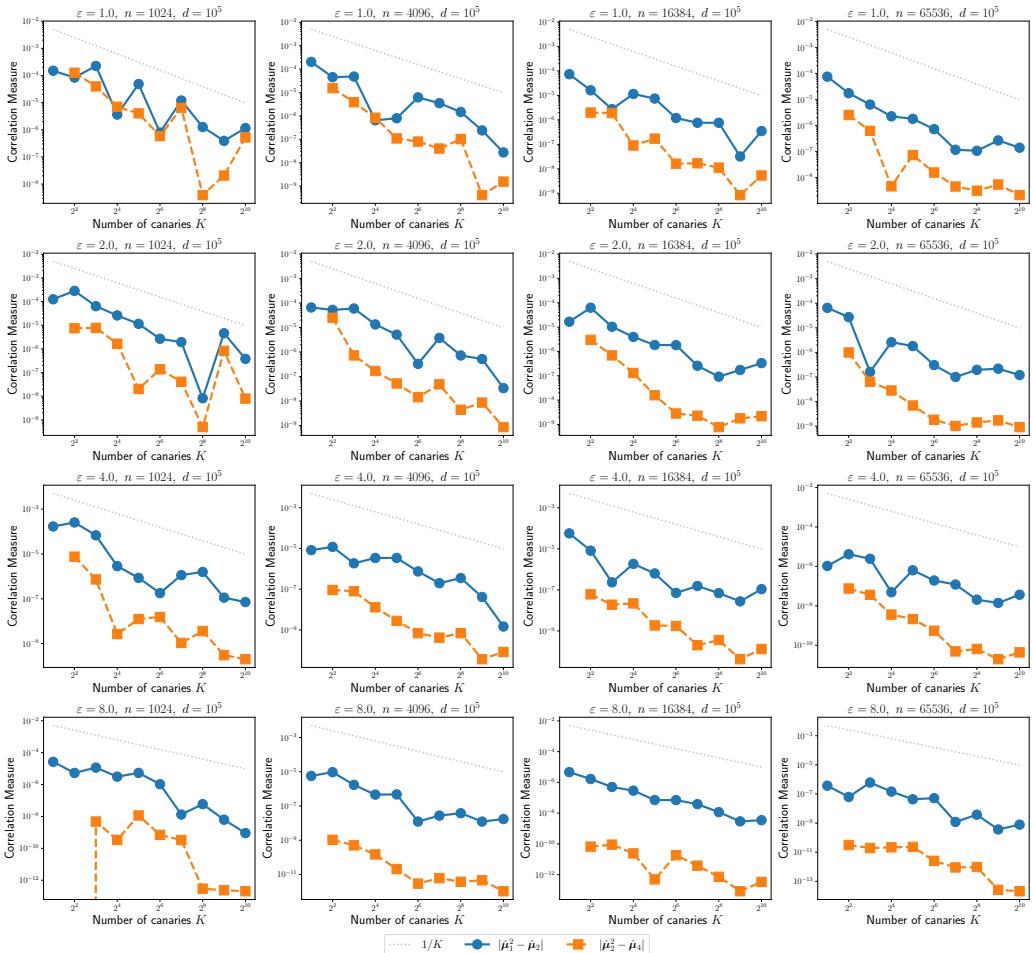

Figure 11: Effect of the number $k$ of canaries on the moment estimates employed by the higher-order Wilson intervals in auditing LiDP.

- Figure 9 shows the effect of varying the number of canaries $k$, similar to Figure 3 (middle).

- Figure 10 shows the effect of varying the data dimension $d$, similar to Figure 3 (right).

- Figure 11 shows the effect of varying the number of canaries $k$ on the moment estimates, similar to Figure 4 (right).

- Figure 12 shows the effect of varying the data dimension $d$ on the moment estimates, similar to Figure 4 (right).

We observe that the insights discussed in §4 hold across a wide range of the parameter values. In addition, we make the following observations.

**The benefit of higher-order confidence estimators.** We see from Figures 8 to 10 that the higher-order Wilson estimators lead to larger relative improvements at smaller $\varepsilon$. On the other hand, they perform similarly at large $\varepsilon$ (e.g., $\varepsilon = 8$) to the lower-order estimators.

**4th-Order Wilson vs. 2nd-Order Wilson.** We note that the 4th-order Wilson interval outperforms the 2nd-order Wilson interval at $\varepsilon = 1$, while the opposite is true at large $\varepsilon = 8$. At intermediate values of $\varepsilon$, both behave very similarly. We suggest the 2nd-order Wilson interval as a default because it is nearly as good as or better than the 4th-order variant across the board, but is easier to implement.

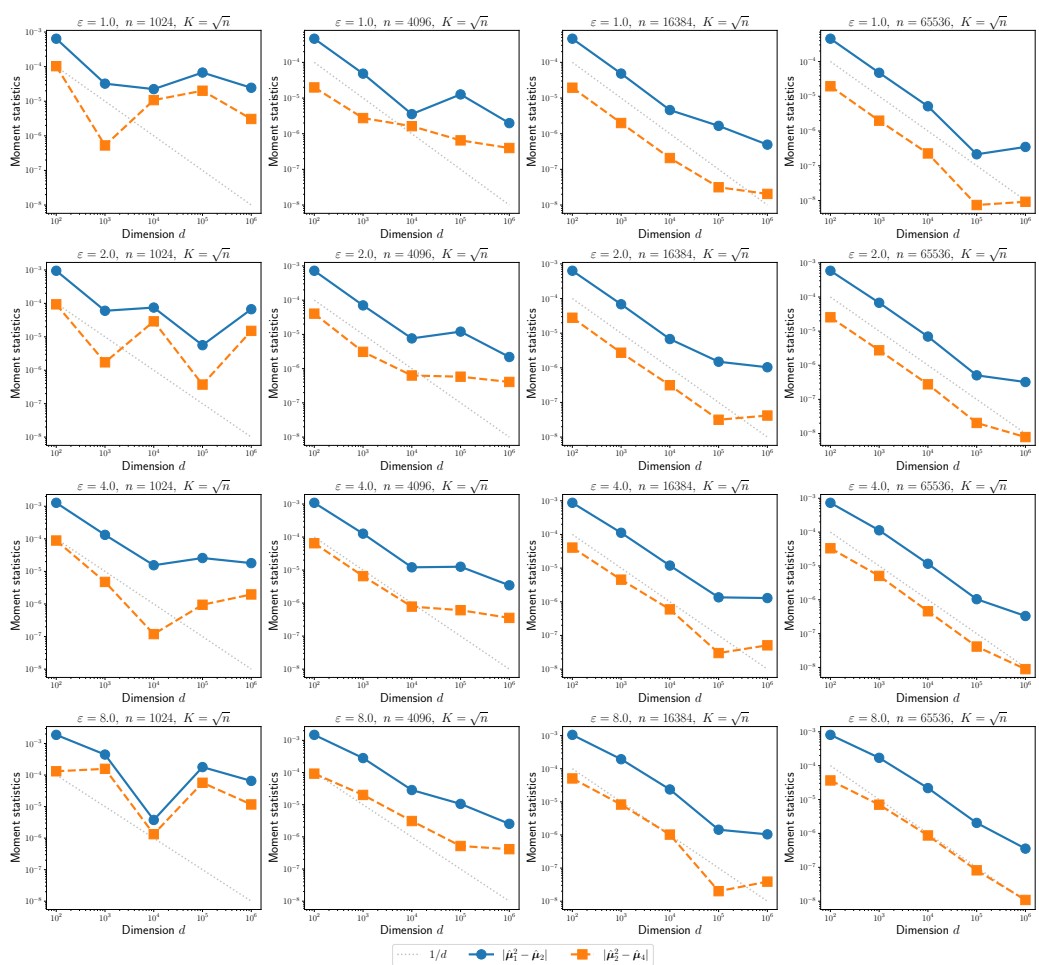

Figure 12: Effect of the data dimension on the moment estimates employed by the higher-order Wilson intervals in auditing LiDP.

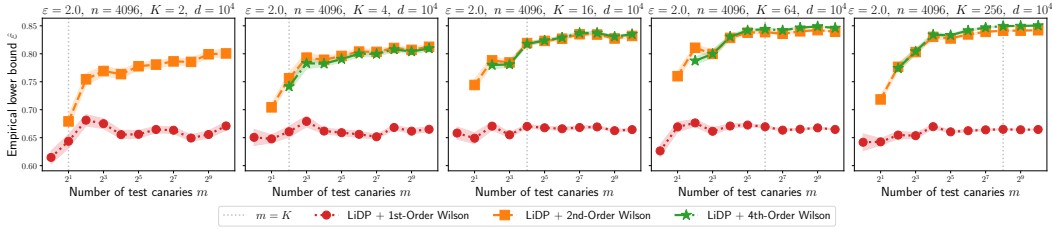

Figure 13: Effect of varying the number $m$ of test canaries with the number $K$ of canaries inserted in the training being fixed.

# F   Experiments: Details and More Results

We describe the detailed experimental setup here.

## F.1   Training Details: Datasets, Models

We consider two datasets, FMNIST and Purchase-100. Both are multiclass classification datasets trained with the cross-entropy loss using stochastic gradient descent (without momentum) for a fixed epoch budget.

- **FMNIST**: FMNIST or FashionMNIST [65] is a classification of $28 \times 28$ grayscale images of various articles of clothing into 10 classes. It contains 60K train images and 10K test images. The dataset is available under the MIT license. We experiment with two models: a linear model and a multi-layer perceptron (MLP) with 2 hidden layers of dimension 256 each. We train each model for 30 epochs with a batch size of 100 and a fixed learning rate of 0.02 for the linear model and 0.01 for the MLP.

- **Purchase-100**: The Purchase dataset is based on Kaggle's "acquire valued shoppers" challenge [19] that records the shopping history of 200K customers. The dataset is available publicly on Kaggle but the owners have not created a license as far as we could tell. We use the preprocessed version of [53][3] where the input is a 600-dimensional binary vector encoding the shopping history. The classes are obtained by grouping the records into 100 clusters using $k$-means. We use a fixed subsample of 20K training points and 5K test points. The model is a MLP with 2 hidden layers of 256 units each. It is trained for 100 epochs with a batch size of 100 and a fixed learning rate of 0.05.

## F.2 DP and Auditing Setup

We train each dataset-model pair with DP-SGD [1]. The noise level is calibrated so that the entire training algorithm satisfies $(\varepsilon, \delta)$-differential privacy with $\varepsilon$ varying and $\delta = 10^{-5}$ fixed across all experiments. We tune the per-example gradient clip norm for each dataset, model, and DP parameter $\varepsilon$ so as to maximize the validation accuracy.

**Auditing Setup.** We follow the LiDP auditing setup described in Appendix B.2. Recall that auditing LiDP with $K = 1$ canaries and corresponds exactly with auditing DP. We train $n \in \{125, 250, 500, 1000\}$ trials, where each trial corresponds to a model trained with $K$ canaries in each run. We try two methods for canary design (as well as their corresponding rejection regions), as discussed in Appendix D: data poisoning and random gradients.

With data poisoning for FMNIST, we define the canary distribution $\mathcal{P}_{\text{canary}}$ as the uniform distribution over the last $p = 284$ principal components (i.e., principal components 500 to 784). For Purchase-100, we use the uniform distribution over the last $p = 300$ principal components (i.e., principal components 300 to 600).

For both settings, we audit *only the final model*, assuming that we do not have access to the intermediate models. This corresponds to the blackbox auditing setting for data poisoning and a graybox setting for random gradient canaries.

## F.3 Miscellaneous Details

**Hardware.** We run each job on an internal compute cluster using only CPUs (i.e., no hardware accelerators such as GPUs were used). Each job was run with 8 CPU cores and 16G memory.

## F.4 Additional Experimental Results

### F.4.1 Comparison to Bayesian Auditing of DP

We compare the proposed LiDP auditing approach with the Bayesian approach of Zanella-Béguelin et al. [69]. This approach constructs a posterior distribution over the true and false positive rates by updating the non-informative Jeffreys priors using empirical observations. They use Bayesian credible intervals (as opposed to confidence intervals in our case) computed using numerical quadrature.

We compare the proposed LiDP auditing with $K = m > 1$ with the Bayesian approach of [69] with $K = m = 1$ in Figure 14 using the code open-sourced by the authors.[4] Owing to the increased runtime in the numerical estimation of the $\varepsilon$ of the Bayesian approach, we tune its threshold $\tau$ to separate the null and alternate hypotheses (cf. line 5 of Algorithm 1) for the Clopper-Pearson interval.

We observe that the LiDP auditing approach generally provides improved estimates when compared to this baseline. Note that the reason behind the improvement for the Bayesian approach of [69] is by capture the joint distribution of the true and false positive rates, while the improvement behind LiDP

---

[3]https://github.com/privacytrustlab/datasets
[4]https://github.com/microsoft/responsible-ai-toolbox-privacy

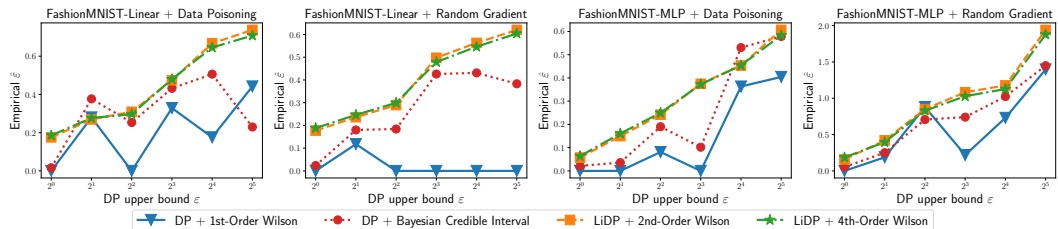

Figure 14: Comparing the proposed LiDP auditing approach to the Bayesian credible interval approach of Zanella-Béguelin et al. [69].

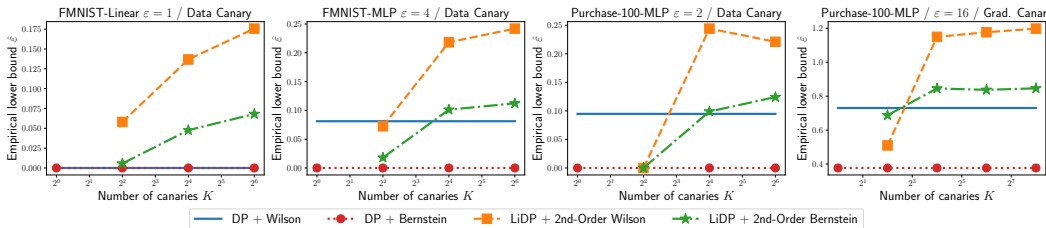

Figure 15: Comparison of higher-order Bernstein and Wilson confidence intervals for LiDP auditing.

auditing is by a clever use of randomized tests. Both sources of improvements are orthogonal to each other. This suggests an interesting future research direction: adapt Bayesian credible intervals for LiDP auditing to get the best of both worlds.

### F.4.2 Comparison of Asymptotic vs. Non-asymptotic Confidence Intervals

We compare the asymptotic Wilson intervals (used primarily in the experiments) to the non-asymptotic Bernstein intervals (used primarily for the theory) in Figure 15. The results show that the Wilson intervals give larger empirical lower bounds in general with all other parameters remaining the same. This is as expected because the Wilson intervals are an asymptotic tightening of the Bernstein intervals as shown in Figure 2.

### F.4.3 Experiments Across Different Parameter Settings

We give the following plots to augment Table 1 and Figure 6:

- Figure 16: plot of the test accuracy upon adding $K$ canaries to the training process.
- Figure 17: experimental results for FMNIST linear model.
- Figure 18: experimental results for FMNIST MLP model.
- Figure 19: experimental results for Purchase MLP model.

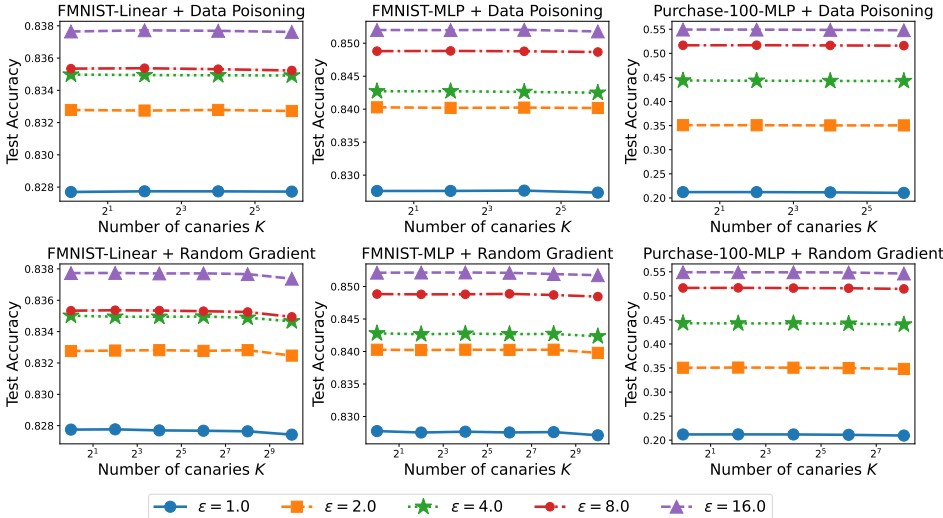

Figure 16: Test accuracy versus the number of canaries $K$. We plot the mean over 1000 training runs (the standard error in under $10^{-5}$). Adding multiple canaries to audit LiDP does not have any impact on the final test accuracy of the model trained with DP.

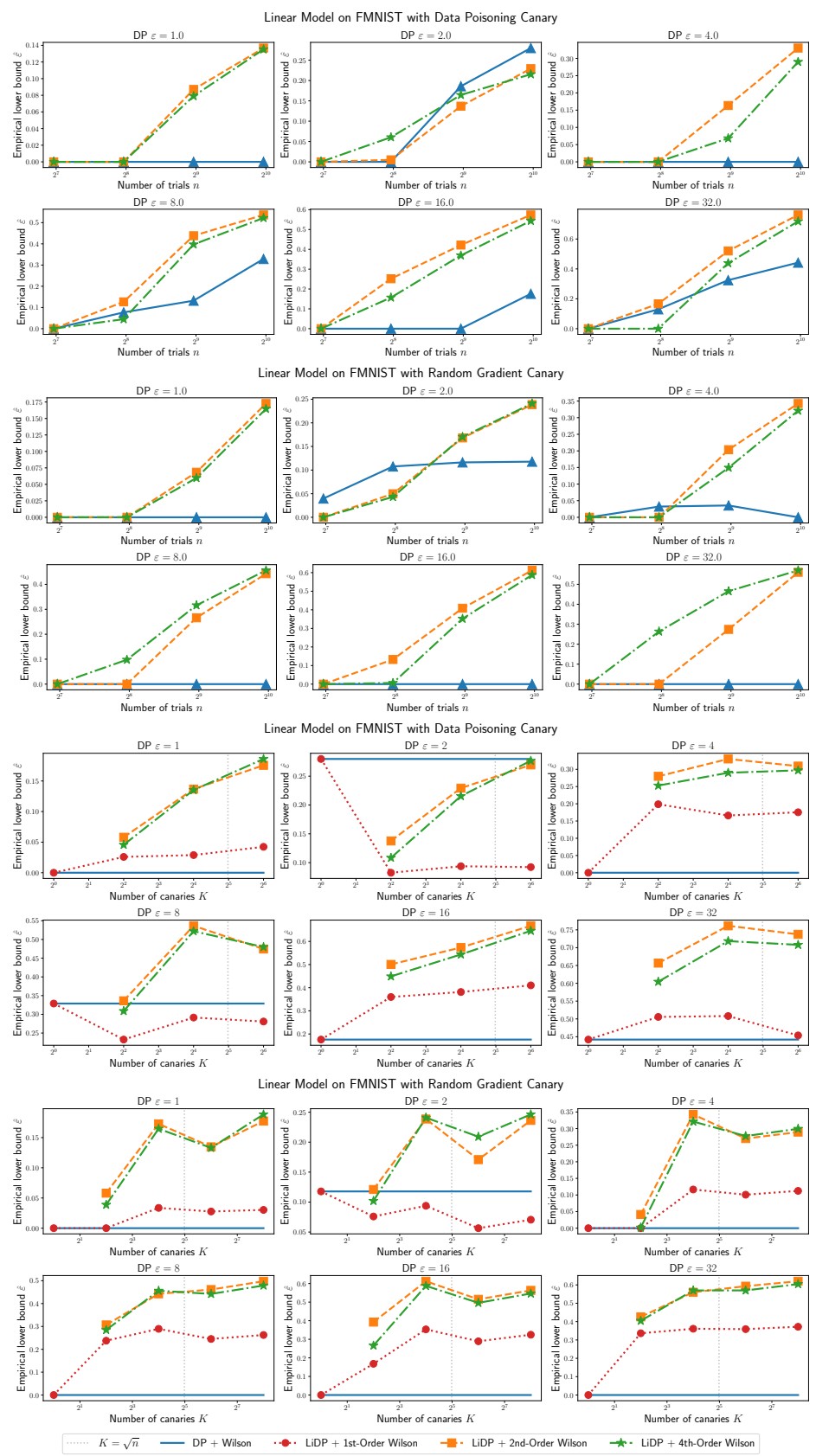

Figure 17: Experimental results for FMNIST linear model (top two: varying $n$, bottom two: varying $K$).

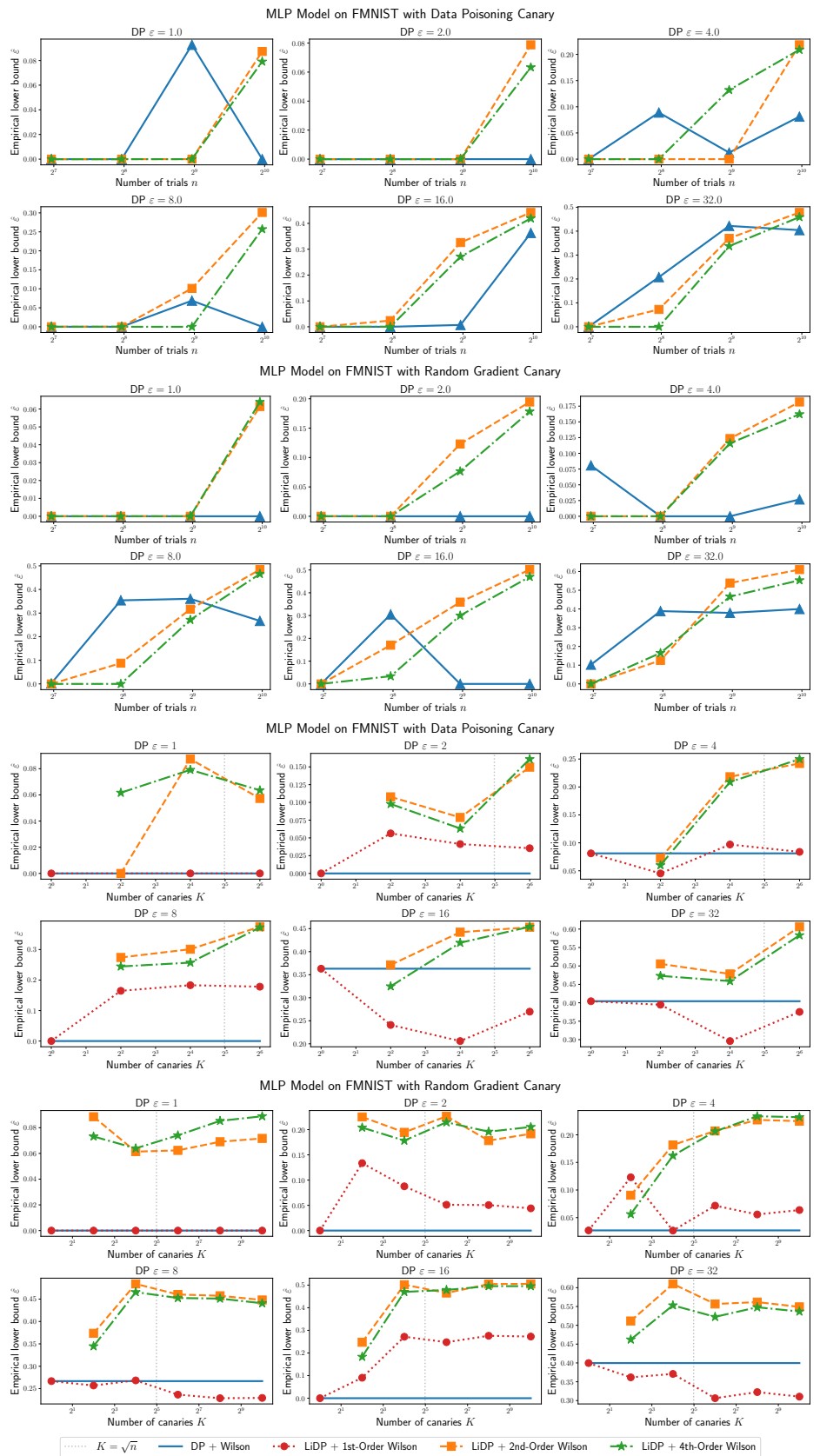

Figure 18: Experimental results for FMNIST MLP model (top two: varying $n$, bottom two: varying $K$).

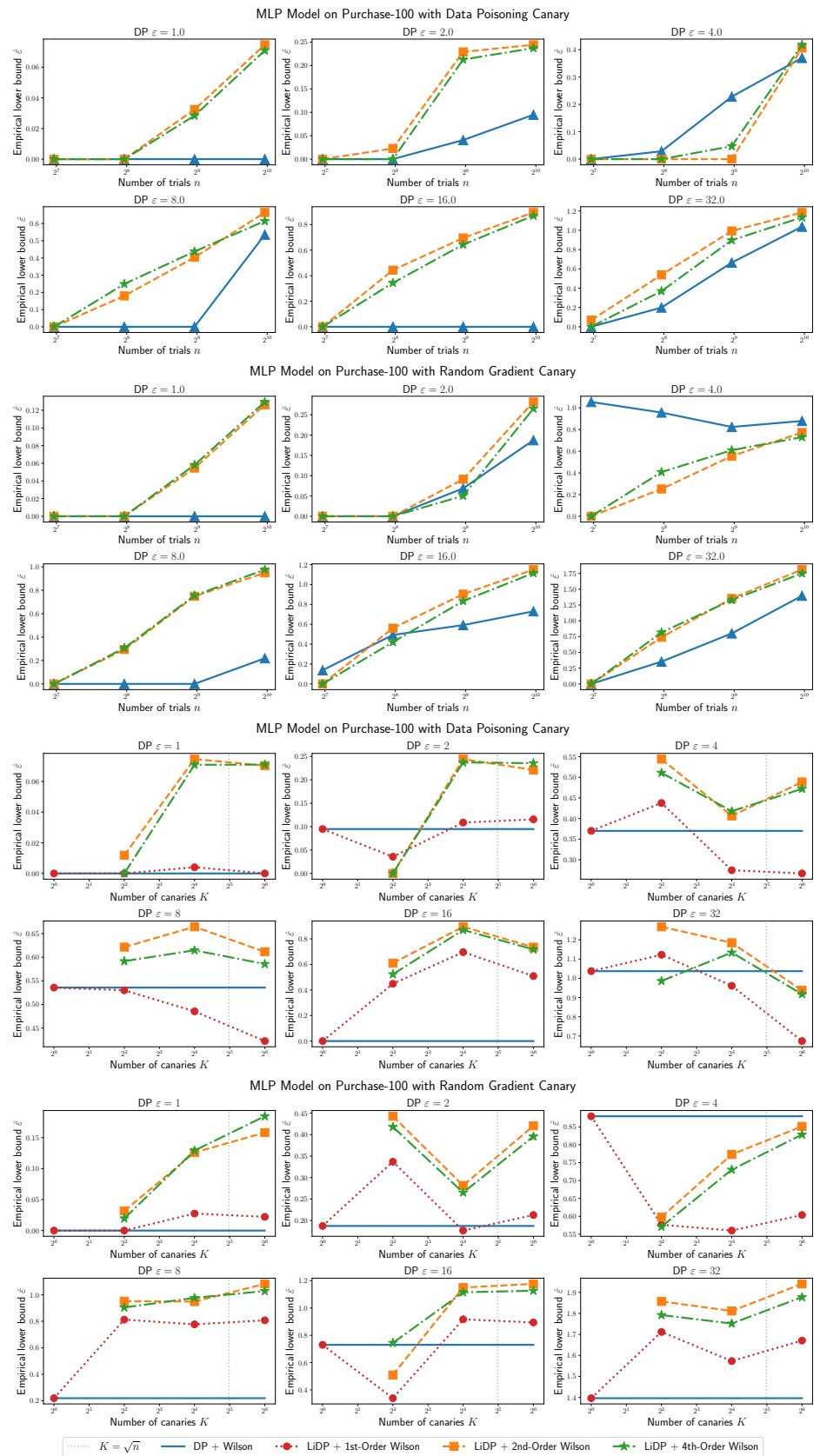

Figure 19: Experimental results for Purchase-100 MLP model (top two: varying $n$, bottom two: varying $K$).

