# OpenReview forum: "Unleashing the Power of Randomization in Auditing Differentially Private ML"
_NeurIPS.cc/2023/Conference — NeurIPS 2023 poster_

### Official Review · Reviewer_W8eW · 2023-06-26

**Soundness:** 3 good
**Presentation:** 3 good
**Contribution:** 2 fair
**Rating:** 5
**Confidence:** 3

**Summary:**

The use of canaries is a very intuitive strategy to evaluate the DP mechanism (which can subsequently attract a broader non-DP specialist audience to this area). The paper is structured well, has comprehensive details on both the method and the corresponding experimental results and shows clear motivation and conclusions. This work presents methodology for auditing of DP trained models through the introduction of Lifted DP and canary-based auditing techniques.

**Strengths:**

The use of canaries is a very intuitive strategy to evaluate the DP mechanism (which can subsequently attract a broader non-DP specialist audience to this area). The paper is structured well, has comprehensive details on both the method and the corresponding experimental results and shows clear motivation and conclusions.

**Weaknesses:**

While the method show promise, the evaluation results are not very representative: FMNIST is a rather small dataset and so is Purchase-100. Given that the models associated with these are also rather small too, I do not see how well can this method scale to larger architectures and more complex datasets (especially as the canary-generation process can heavily depend on it).

It is not very clear to me what the advantage is over the work of https://arxiv.org/abs/2305.08846. Here model auditing can be performed in a single training run, rendering other methods of auditing significantly less efficient. I would like the authors to comment on why their work is a substantial contribution given the existence of this method.

**Questions:**

What is the dependency with epsilon level? There does not seem to be a clear trend of how well the method performs with more noise being added. I would like the authors to discuss this further and hypothesise what the trend is and why.

Could the authors elaborate on what they mean by ‘poisoning’ canaries? What is the methodology like on non-image modalities?

**Limitations:**

I described most of the limitations in the weaknesses section, but to summarise: A) it is not clear how the method performs in non-trivial settings (e.g. ImageNet), B) the improvements compared to the work I linked above are not clear.
Overall, this work shows promise, but I would like the authors to clarify the points I raised above before I can recommend acceptance.

---

> ### Author Rebuttal · Authors · 2023-08-09
>
>  We thank the reviewer for the questions.
>
> > **I would like the authors to comment on why their work is a substantial contribution given the existence of the method [of Steinke et al.]**
>
> We make **4 key contributions**: (1) the framework of lifted DP, (2) a novel randomized leave-one-out hypothesis test for auditing lifted DP, (3) adaptive confidence intervals, (4) detailed numerical results of the frameworks, including the bias-variance trade-offs it involves.
>
> The independent and concurrent work of Steinke et al. also considers auditing with randomized canaries that are Poisson-sampled, i.e., each canary is included or excluded independently with equal probability. **Their recipe is different from ours**: it involves computing an empirical lower bound by comparing the accuracy (rather than the full confusion matrix as in our case) from the possibly dependent guesses with the worst-case randomized response mechanism. This allows them to use multiple dependent observations from a single trial. Our **confidence intervals** are adaptive to the level of correlation between the canaries, while theirs are non-adaptive and worst-case.
>
> Our work gives a **deeper analysis of the bias-variance tradeoff** from utilizing randomized canaries. Further, we develop two **novel technical tools**, LiDP and adaptive XBern confidence intervals, which could be of general interest to the community.
>
> We are excited about the possibilities of combining our adaptive confidence intervals (the key to our improvements) with the framework of Steinke et al. (the key to their improvements) to get the best of both worlds. We leave this for future work.
>
> Finally, we note that the work of Steinke et al. was submitted to arXiv on 15 May 2023, i.e. **2 days before the NeurIPS deadline** on 17 May 2023). Per the **[NeurIPS policy](https://nips.cc/Conferences/2023/CallForPapers)**, "[a] submission will not be rejected on the basis of the comparison to contemporaneous work". We request the reviewer assess the merits of our substantial theoretical and methodological contributions independently.
>
>
> > **The evaluation results are not very representative… larger architectures and more complex datasets ... non-trivial settings (e.g. ImageNet)**
>
> We opted for small datasets and models so we could test our theoretical predictions *comprehensively* and compare them with *previous* baselines in the literature. We run **150K experiments for each dataset-model pair** (6 epsilons * 6 values of K * 2 types of canaries * 2000 seeds per setting), which would be infeasible at larger scales. Small datasets and models have also been the norm for the privacy auditing literature, see e.g. [Jagielski et al. NeurIPS 2020](https://arxiv.org/abs/2006.07709), [Nasr et al. IEEE S&P 2021](https://arxiv.org/abs/2101.04535), [Lu et al. NeurIPS 2022](https://arxiv.org/abs/2210.08643).
>
> We note that the purpose of auditing is to check for the sanity of the training algorithm, which (in principle) is independent of the model size and the dataset size. One could in practice use a smaller model and a smaller dataset to audit an algorithm, and, if it passes, deploy a larger model on a larger dataset. We agree that this is not ideal, but perhaps is a practical compromise to speed up the adoption of these tools.
>
> Further, demonstrating rigorous privacy auditing at ImageNet scale for the *first time* would in itself be a *significant milestone* in the field. This is outside the scope of our paper, whose key contributions are methodological, foundational, and theoretical.
>
> > **Question: What is the dependency with epsilon level?**
>
> Auditing DP grows exponentially harder with increasing epsilon. Consider for instance, distinguishing between randomized response $\text{RR}(\infty)$ vs. $\text{RR}(\varepsilon)$, which is equivalent to distinguishing between $\text{Bern}(1)$ vs. $\text{Bern}(\sigma(\varepsilon))$ where $\sigma(\cdot)$ is the sigmoid function. By known lower bounds in hypothesis testing, this requires at least $n \ge O(1/(1 - \sigma(\varepsilon))^2) \approx e^{2\varepsilon}$ samples, which is exponential in $\varepsilon$. Thus, we can only effectively audit smaller $\varepsilon$.
>
> Empirically, we see from **Figure 5 of the rebuttal PDF** that LiDP auditing greatly outperforms the baseline at small $\varepsilon$ (by a multiplicative factor of 2x to 4x). At large $\varepsilon$, both methods are constrained by the fundamental difficulty described above and perform similarly.
>
>
> > **Why do we refer to ‘poisoning’ canaries?**
>
> The phrase “poisoning” comes from the literature on adversarial machine learning where an adversary can “poison” the model training by adding specific (outlier) data points. Much of the prior work on designing canaries for auditing is inspired by this literature. See e.g. [Jagielski et al. (NeurIPS 2020)](https://arxiv.org/abs/2006.07709) for more historical context behind this nomenclature.
>
> > **What is the methodology (to design data poisoning canaries) on non-image modalities? + Canary designs for complex datasets**
>
> We run our experiments on image and tabular data. Prior work has considered designing canaries for text/speech in the context of memorization analysis (rather than DP auditing). For the text domain, prevailing approaches insert random sequences possibly based on templates as canaries [e.g. [Carlini et al.](https://arxiv.org/abs/1802.08232), [Mireshghallah et al.](https://arxiv.org/abs/2103.07567)]. For the speech modality, prior work uses a text-to-speech engine to convert such text canaries to the speech modality [[Jagielski et al.](https://arxiv.org/abs/2207.00099) p.16].
>
> Our framework of LiDP auditing is flexible enough to incorporate recent advances in designing canaries for more complex datasets by lifting as illustrated in Section 5 and Appendix D. The aforementioned strategies can be lifted to audit LiDP as they are random by construction. This would be an interesting exploration for future work.

---

> > ### Comment · Reviewer_W8eW · 2023-08-14
> > **Response to the rebuttal**
> >
> > I would like to thank the authors for responding to my comments.
> >
> > A lot of their responses have really clarified certain aspects of the work (particularly the part on different levels of epsilon and the comparison to the method I have linked above). You are right in saying that this work was overlapping with this submission, so I will remove this issue from consideration.
> >
> > I am still not entirely happy with the comment on larger datasets, as in my mind, any canary-based method could suffer significantly from more complex datasets, potentially rendering these methods infeasible.
> >
> > I will, nonetheless, raise the score in light of my first comment, but would ideally still like an extended discussion (if not experimental evidence) on how well the method scales with some examples. Score has been raised accordingly and happy to update it further if this comment is addressed in detail.

---

> > > ### Author Response · Authors · 2023-08-14
> > > **Thank you for your constructive feedback.**
> > >
> > > We would like to thank the reviewer for their constructive feedback. We agree that as the dataset (and the task) gets larger and more complex, the kind of canary tests we do might become weaker, thus making the lower bound smaller. It is an important challenge not just for our paper but for the community of researchers working on privacy auditing. We will add such discussion in the revision. We will get back with a more detailed response soon.

---

> > > ### Author Response · Authors · 2023-08-15
> > > **Thank you for the response!**
> > >
> > > We thank the reviewer for the response!
> > >
> > > There are two major research questions in privacy auditing: (i) designing stronger canaries, and (ii) developing tighter confidence intervals. One needs both to achieve a tight lower bound on epsilon. In this manuscript, we focus on the latter question.
> > >
> > > Our approach inherits the strengths (and weaknesses) of existing canary designs but provides a better way to reduce the width of the confidence interval with a principled use of randomness. We try to make it clear that this paper does *not* attempt to design stronger canaries. We note that the gain of our approach still depends on the canary design via how correlated the randomly drawn canaries are. That said, there is no reason to believe that the canaries will have a larger correlation when applied to complex tasks (and datasets).
> > >
> > > At the same time, we agree with the reviewer that if one uses weak canaries, any canary-based method will fail. Ours is no exception here. However, this issue is orthogonal to the main contributions of our paper. In our view, it is outside the scope of a theoretical paper to (i) check the strengths of the canaries for larger and more complex datasets, and (ii) design stronger canaries for such tasks. These are timely and important research directions for the privacy auditing community. We will add a discussion on this subject. Thank you for an excellent suggestion!

---

### Official Review · Reviewer_2DXh · 2023-07-05

**Soundness:** 4 excellent
**Presentation:** 4 excellent
**Contribution:** 4 excellent
**Rating:** 8
**Confidence:** 4

**Summary:**

The authors propose a principled improvement over existing privacy auditing techniques. Privacy auditing through the binary hypothesis testing formulation implies gathering samples on the success of the adversary when trying to correctly guess the membership game against a mechanism $M$. All auditing techniques involve switching to a high-probability argument via confidence intervals to be able to provide lower bounds. These CIs converge with a rate of $O(\frac{1}{\sqrt{N}})$, where N is the number of auditing samples. The goal of this paper is to improve this convergence rate by testing multiple (possibly correlated) samples at a time, providing an improvement for the convergence of up to $O(\frac{1}{\sqrt{NK}})$, where K is the number of inserted canaries. Their proposed technique is as follows: 1. Run the mechanism $M$ on $D \cup ( c_1, .., c_K ) $ and $D \cup (c_1, .., c_{K-1}) $, where the canaries are iid samples drawn from a fixed canary distribution. Through a clever randomisation argument, using the property that the canaries are iid from the same distribution, the authors propose testing multiple iid canaries by testing other iid canaries by exchanging the missing $c_K$ with other iid canaries. By doing this, the authors can test $K$ test statistics for the alternative hypothesis and $M$ test statistics for the null hypothesis. Now, the goal is to integrate this new information into the confidence interval. The authors do not make an independence assumption as in previous work and derive a tailored confidence interval that accounts for the possible correlation between the K tests. Their analysis shows how their confidence interval converges with a rate of $O(\frac{1}{\sqrt{N
}})$ when the statistics are correlated to a rate of $O(\frac{1}{\sqrt{NK}})$ when the statistics are independent.

**Strengths:**

* The paper has a nice structure, it solves a clearly stated problem known in the privacy auditing community, for which they introduce all the needed formalism and they make the necessary experiments to back up their results.
* The authors propose an algorithm that could be used as a drop-in replacement for other auditing techniques.
* Formally accounting for the correlation between hypotheses fills an existing gap in auditing literature when multiple samples are used for auditing purposes. All the literature I am aware of makes an independence assumption, which might be strong in some cases. This work recovers that case but accounts for others as well, framing and formalising in a principled way the multisample testing problem.


**Weaknesses:**

* in the experimental section, the authors show a bias/variance tradeoff when auditing a mechanism M that computes a noisy sum/mean. The same argument does not hold against differentially private machine learning models, and it would be interesting to show if the proposed technique can achieve tightness when auditing a (possibly simple) machine learning pipeline.


**Questions:**

* The proposed auditing technique is not tailored for the Gaussian mechanism but for any mechanism $M$. For example, one could audit a Laplace Mechanism or a Subsampled Gaussian Mechanism with it. I think the authors could stress this out for the reader, as improving auditing sample complexity in the general case is a truly desirable property, and this work does that. Would the authors want to add some details or employ their technique to audit more diverse mechanisms?
* The bias/variance argument in section 4 holds against a mechanism that averages over a list of gradients, as it is easy to observe how inserting noisy samples will yield privacy amplification in that scenario. Yet, when we audit a machine learning model with input poisoning (so, actual samples need to be inserted, not the gradients), the relationship is not so clear anymore. How many sample canaries could we use for auditing purposes against a machine-learning model? Is there any scenario in which we can hope to audit correctly by training only two models, for example? (n=1, k=?) What is the performance in that scenario?


**Limitations:**

-

---

> ### Author Rebuttal · Authors · 2023-08-09
>
> We thank the reviewer for a detailed review and thought-provoking questions.
>
> > **Bias-variance for ML pipelines + auditing with input poisoning**
>
> We agree with the reviewer that for the Gaussian mechanism, we have all the information to (numerically) compute the bias and variance exactly for our estimator of the TPR and FPR (and consequently $\varepsilon$). This clearly shows the bias-variance tradeoff that we discuss in the paper.  For an ML pipeline, things are more complicated for obvious reasons as the reviewer insightfully pointed out, but we still believe that a similar bias-variance trade-off holds.
>
> **Bias increases (i.e., lower bound on $\varepsilon$ decreases) with larger $K$**: This is challenging to show formally, but intuitively we believe this is true for the following reason. Let us consider the actual theoretical $\varepsilon$ of the mechanism with $K$ random canaries. The privacy (treating the canary $c_K$ are the one that is absent/present in neighboring datasets) is governed by the pair of distribution $P_{A,c_1^K}(\theta|c_K)$ and $P_{A,c_1^{K-1}}(\theta)$, where $A$ accounts for the randomness internal to the training mechanism and $c_1^K$ accounts for the randomness in the canaries 1 to $K$.
>
> We argue that the bias appears because the theoretical $\varepsilon$ for $K>1$ should be smaller than that of $K=1$. This is suggested by the fact that the pair $P_{A,c_1^K}(\theta|c_K)$ and $P_{A,c_1^{K-1}}(\theta)$ when $K>1$ has more *randomness* coming from the other canaries $c_1^{K-1}$ that act as if they are part of the privacy mechanism, and hence gives a higher variance output (which is more private, and hence smaller $\varepsilon$). We agree that this is hand-wavy and non-rigorous, and probably can only be confirmed empirically.
>
> **Variance decreases with larger $K$**: This refers to the variance of our estimators on TPR and FPR here, and the dominant term in the derivation (for example for the Bernstein bound) decreases with larger $K$. This is regardless of the mechanism that is being considered and should hold for general ML pipelines. However, again the lower order terms can increase marginally with $K$, so for this to be absolutely true, we need large enough $n$ and small enough correlation between the canaries.
>
> We will add more details about this to the paper.
>
> > **Would the authors want to add some details or employ their technique to audit more diverse mechanisms (Laplace, etc.)?**
>
> We thank the reviewer for the fantastic suggestion. Indeed, our recipe can be used to audit any mechanism including a general class of mechanisms built from a composition of additive noise mechanisms and subsampling. Note that the DP-SGD we audit in Section 6 includes sub-sampling and amplification by sampling in the mechanism and privacy accountant (for the theoretical upper bound).
>
> In the original submission, we only tried DP-SGD due to its overwhelming popularity in practice. Here, as a proof of concept, we additionally audit a Laplace mechanism using canaries uniformly from the $L_1$ sphere. The results are qualitatively similar to those of the Gaussian case with a near 8x gain in the sample complexity. We refer to **Figure 4 of the rebuttal PDF** for detailed results.
>
> > **How many sample canaries could we use for auditing purposes against a machine-learning model?**
>
> In our experiments with real data training, we observed that our lower bound improves with $K$ up to $K=64$ for Purchase data and $K=256$ for FashionMNIST. Afterward, we believe the lower bounds should start to get worse (get smaller) as $K$ increases. Choosing the right $K$ in practice is a difficult task, although using $K=\sqrt{N}$ seems to give reasonable results.
>
> > **Is there any scenario in which we can hope to audit correctly by training only two models, for example? (n=1, k=?) What is the performance in that scenario?**
>
> Great question! Our method currently requires $n \ge K^{\ell / (\ell - l)}$ for an $\ell$th order confidence estimator (Proposition 4). No lower bounds are known here and whether this can be reduced to $n=1$ is an open question.
>
> The independent and concurrent work of [Steinke et al.](https://arxiv.org/abs/2305.08846) proposes privacy auditing with $n=1$ (this paper appeared on arXiv only 2 days before the NeurIPS deadline). It is not yet clear how their method compares to ours as the experimental settings are not comparable. There is a potential to harness the benefits of both approaches, which we leave as a future research direction.

---

> > ### Comment · Reviewer_2DXh · 2023-08-14
> >
> > Thank you for your great and detailed answers, looking forward to the discussions with other reviewers!

---

### Official Review · Reviewer_jDkd · 2023-07-07

**Soundness:** 4 excellent
**Presentation:** 3 good
**Contribution:** 3 good
**Rating:** 7
**Confidence:** 4

**Summary:**

This work studies the sample complexity (in terms of sampled models) of auditing differential privacy. It proposes a new approach to usage of canaries in auditing via hypothesis test, breaking an existing sample complexity barrier by making use of multiple, randomized canaries. These randomized canaries are used to audit a new, but equivalent formulation of approximate differential privacy called lifted DP, which states a privacy condition for neighboring datasets and rejection sets sampled from a distribution. Sampled models are reused to compute test statistics for each of the multiple canaries included in the dataset, and confidence intervals for the average test statistic are computed using techniques introduced in the paper, tailored to the distribution of test statistics produced by the new auditing method. Empirical evaluation of the proposed method is also given.

**Strengths:**

The paper makes progress in improving the sample complexity of DP auditing by introducing new techniques that should be compatible with future progress in strengthening canaries. The usage of multiple, randomized canaries is innovative and novel tools were used to adapt their usage to existing hypothesis testing frameworks for auditing. The empirical results in Section 4 and 6 were quite helpful in interpreting the expected sample complexity improvements from various choices of K and order of confidence interval.

**Weaknesses:**

The sample complexity improvements seem notable, but I'm curious about how large of a bottleneck sample complexity is to practical auditing efforts. I'd expect it to be a significant obstacle, but additional discussion of the extent to which sample complexity impedes privacy auditing could help further motivate this work.

Notes:
pg 5 - "in practice, it depends on how correlated the K test are"
In the caption of figure 3 - "provides significant gain in the require number of trials to achieve a desired level of lower bound"
pg 9 - "For deployment in production, it worth further studying approaches with minimal computational overhead"

**Questions:**

See comments in weaknesses.

**Limitations:**

Yes, the authors adequately addressed potential societal impact.

---

> ### Author Rebuttal · Authors · 2023-08-09
>
>  We thank the reviewer for a constructive review.
>
> **Additional discussion on the bottleneck of sample complexity for privacy auditing:**
>
> Auditing DP is fundamentally constrained by the sample size $n$, where each sample corresponds to one trained model. Rigorous lower bounds on privacy require $1/\sqrt{n}$ confidence intervals. To get convincing results, [Tramer et al.](https://arxiv.org/pdf/2202.12219.pdf) train $n=10^5$ models, but auditing generally requires $n=O(10^3)$ [[Jagielski et al.](https://proceedings.neurips.cc/paper_files/paper/2020/file/fc4ddc15f9f4b4b06ef7844d6bb53abf-Paper.pdf)]. This makes it unusable for all but the smallest models and datasets. Past approaches use heuristics without rigorous justification to avoid this large computational cost, e.g. [[Zanella-Béguelin et al.](https://arxiv.org/pdf/2206.05199)]. We will add this quantitative discussion in the motivation section.
>
> We introduce LiDP, which is equivalent to regular DP, but lets us use randomized canaries. We present a novel recipe based on a randomized leave-one-out hypothesis test to audit LiDP. We propose adaptive confidence intervals to leverage multiple correlated observations from each experiment. Altogether, we obtain the same rigorous lower bounds as the baselines with up to 16x smaller sample size for synthetic simulations and up to 5x for real data experiments.
>
> This directly translates into a 5x gain in the run-time of auditing, enabling the use of rigorous privacy auditing with a reduced computational cost. This, we believe, will speed up and broaden the adoption of these important tools for auditing private mechanisms. Further, the novel technical tools we develop in this paper — LiDP and adaptive XBern confidence intervals — might be of general interest beyond auditing.

---

> > ### Comment · Reviewer_jDkd · 2023-08-18
> >
> > I thank the authors for their response detailing to practical implications of their work. I will amend my confidence upwards.

---

### Official Review · Reviewer_1JBM · 2023-07-19

**Soundness:** 3 good
**Presentation:** 4 excellent
**Contribution:** 3 good
**Rating:** 7
**Confidence:** 5

**Summary:**

This paper presents a methodology for calculating lower bounds for the parameter $\\varepsilon$ of an $(\\varepsilon,\\delta)$-differentially private (DP) mechanism. When applied to machine learning algorithms, the methodology can be used to audit the privacy guarantees of popular training algorithms such as DP-SGD.

The paper builds on existing methods that audit DP guarantees by testing the presence of a canary in the input of the mechanism. Given an $(\\varepsilon,\\delta)$-DP mechanism $\\mathcal{A}$, two neighboring datasets $D_0$ and $D_1 = D_0 \\cup \\{c\\}$ and a measurable set $R$, a lower bound for $\\varepsilon$ is given by

$$
   \\varepsilon \\geq
   \\log (\\mathbb{P}(\\mathcal{A}(D_1) \\in R) - \\delta) -
   \\log \\mathbb{P}(\\mathcal{A}(D_0) \\in R) .
$$

A lower bound for $\\varepsilon$ can be derived from a lower bound $\\underline{\\boldsymbol{p}}_1 \\leq \\mathbb{P}(\\mathcal{A}(D_1) \\in R)$ and an upper bound $\\overline{\\boldsymbol{p}}_0 \\geq \\mathbb{P}(\\mathcal{A}(D_0) \\in R)$ estimated from samples. This lower bound can be made tighter by choosing a canary $c$ whose presence in the training dataset can be easily tested with a rejection set $R$. However, using Bernoulli confidence intervals computed from $n$ samples, the bounds $\\underline{\\boldsymbol{p}}_1, \\overline{\\boldsymbol{p}}_0$ are loose by a factor of $1/\\sqrt{n}$. Since each sample requires evaluating $\\mathcal{A}$ on a different dataset $D_0$, decreasing this factor is expensive.

The key idea of the paper is to instead audit a probabilistic lifting of DP (LiDP) using counterexamples sampled from a joint distribution $( \\boldsymbol{D_0},  \\boldsymbol{D_1}, \\boldsymbol{R})$. While entirely equivalent to DP, thanks to the exchangeability of random i.i.d canaries  LiDP allows reusing samples $\\mathcal{A}(D \\cup \\{c_1,\\ldots,c_K\\})$ and $\\mathcal{A}(D \\cup \\{c_1,\\ldots,c_{K-1}\\})$ to gather multiple correlated test statistics $\\boldsymbol{x}_k$, one for each canary $c_k$. A lower bound for $\\varepsilon$ can be then calculated similarly as for DP from bounds of the mean of these statistics. The paper analyzes the correlation between the statistics to derive higher-order exact (Bernstein) and asymptotic (Wilson) confidence intervals. Importantly, the authors prove that $\\ell$-th order bounds are loose by a factor of $n^{(1-2\\ell)/(2\\ell)}$ when choosing $K = \\lceil n^{(\\ell-1)/\\ell} \\rceil$ and so e.g. second-order bounds reduce the number of samples required to attain a given confidence because their looseness decreases as $1/n^{3/2}$ rather than $1/n^{1/2}$ for first-order bounds (and previous methods auditing DP).

The authors evaluate the methodology on a Gaussian mechanism showing a 4x gain in sample efficiency when using second-order confidence intervals with $n=1000$ compared to a baseline using Bernoulli confidence intervals. They also evaluate the method on linear and 2-layer MLP classifiers trained with DP-SGD on FMNIST and Purchase-100 using either random or clipping-aware poisoned canaries [31]. This evaluation shows an average improvement in sample efficiency of up to 3x also with $n=1000$.

The supplemental material includes proofs of the results in the paper, algorithmic descriptions of the methodology, a derivation of asymptotic Wilson intervals, and comprehensive ablation studies varying the privacy budget $\\varepsilon$, the number of canaries $K$, samples $n$, and dimension (for the Gaussian mechanism).


**Strengths:**

1. A novel method for auditing differential privacy that can be combined with existing canary design strategies and improve sample efficiency.

2. First formal analysis of previously used heuristics reusing trained models to obtain multiple samples.

3. Technically solid theoretical foundations. Detailed proofs. Derivation of exact and asymptotic confidence intervals.

4. Great high-level intuition for why the method improves sample efficiency and the reasons for the bias/variance trade-off in selecting the number of canaries to use.

5. Extensive algorithmic descriptions of the method in the supplemental material that make the paper fairly self-contained and enable reproducibility. Authors promise to open-source their code to replicate results.

**Weaknesses:**

1. The evaluation on ML scenarios uses very simple models: a linear model and MLPs with 2 hidden layers with 256 units (269k and 245k parameters for FMNIST and Purchase-100, respectively).

2. Evaluation baselines are limited to approaches using Bernoulli confidence intervals and not to more recent approaches using credible regions [66] which also claim improved sample efficiency.


**Questions:**

1. Using the form of Bernstein's bound I am familiar with (Dudley, Richard M. Uniform central limit theorems. Vol. 142. CUP, 2014) would give a slightly tighter bound than in Equation (6). The bound in the paper seems to use $\\sqrt{a+b} \\leq \\sqrt{a} + \\sqrt{b}$ to simplify it, making it looser. Can you clarify whether you use this simplification, or else provide a detailed derivation?

2. In your evaluation, do you set $m = K$ in Algorithm 1?

3. There seems to be a qualitative difference between $\\underline{\\boldsymbol{p}}_1$ and $\\overline{\\boldsymbol{p}}_0$ in Algorithm 1 that is not discussed in the paper. The former is calculated from statistics $\\boldsymbol{x}^{(i)}$ using canaries $c_1,\\ldots,c_K$ inserted into the training set while the latter is calculated from statistics $\\boldsymbol{y}^{(i)}$ depending on canaries $c'_1,\\ldots,c'_m$ not in the training set that are independent of the output of the mechanism (the statistics themselves are correlated because they all use the same model). The bias-variance tradeoffs that you discuss in §4 seem to apply only to $c_k$. Are there tradeoffs that make increasing the number of out-canaries $m$ not always beneficial?

### Details

- The paper presents exact Bernstein confidence intervals but then uses asymptotic Wilson intervals exclusively in the evaluation. The two are only compared in Figure 2. I would have liked to see an additional comparison in at least some selected empirical results for bounds on $\varepsilon$.

- In Eq.1 $D,D'$ should be $D_0,D_1$. This has been fixed in the PDF in the supplemental material.

- l.119: "the canary has the freedom" => "the **adversary** has the freedom"?

- l.197: "applying probabilistic method" => "applying **a heuristic** method"?

- Figure 1 is not referenced. I think it is supposed to support the analysis in the paragraph starting at line 271.

- In Figure 3 caption: "require number" => "required number"

- l.243: "find §4 that" => "find **in** §4 that"

- l.346: "it worth" => "it **is** worth"

- l.710: $k$ => $K$

- In Algorithm 3 in steps 4 and 5, and in Algorithm 4 in steps 5 and 6, I believe that there is a missing $\frac{K-1}{K}$ multiplying $\overline{\boldsymbol{\mu}}_2$ as in Equation (17).

**Limitations:**

The authors discuss some limitations of the auditing methodology throughout the paper and discuss the need to balance the tightness/computation trade-off in practice.

It could be a good idea to complement this with a discussion of how the lower bounds for $\\varepsilon$ depend on the canary design and detection strategy, how to interpret the gap between lower bounds and the theoretical guarantee of a mechanism, and the impact of deciding whether a mechanism provides enough privacy based on a lower bound alone.

I also believe that given the size and variety of the ML models used in the evaluation, the paper should discuss how the results may extrapolate to other architectures and larger number of parameters. Figure 4 (right) shows the method may benefit as the dimension increases; it would be great if this effect could be backed with experimental results on ML models.

---

> ### Author Rebuttal · Authors · 2023-08-09
>
> We thank the reviewer for a thorough review and thought-provoking questions!
>
>
> > **Weakness 1: ... very simple models and small datasets ...**
>
> These are standard benchmarks in privacy auditing literature, due to computational limitations. We refer to our response at the end of this rebuttal for a discussion. We note that the purpose of auditing is to check for the sanity of the training algorithm, which (in principle) is independent of the model size and the dataset size. In practice, one could use a smaller model and a smaller dataset to audit an algorithm, and, if it passes, deploy a larger model on a larger dataset. We agree that this is not ideal, but perhaps is a practical compromise to speed up the adoption of these tools.
>
> > **Weakness 2: Evaluation with Bayesian credible intervals as baselines**
>
> Following the reviewer’s suggestion, we obtained the lower Bayesian credible interval from the approach of [Zanella-Béguelin et al.](https://arxiv.org/pdf/2206.05199.pdf) Our approach generally provides _improved estimates_ when compared to this baseline. Please see **Figure 1 of the rebuttal PDF** for detailed results.
>
> This suggests an interesting future research direction: adapt Bayesian credible intervals for LiDP auditing to get the best of both worlds.
>
> > **Question 1: Bernstein Inequality**
>
> We use the analytically simpler bound since our purpose of deriving the Bernstein bound is to understand the tradeoff and their asymptotic dependence. The standard form of Bernstein’s inequality states that the empirical mean $\hat\mu_n$ from $n$ samples of i.i.d. $b$-bounded random variables with mean $\mu$, variance $\sigma^2$ is bounded with probability at least $1-\beta$ as
> $$
> \mu - \hat \mu_n \le \sqrt{\frac{2\sigma^2}{n} \log\frac{1}{\beta} + \frac{b^2}{9n^2} \log^2 \frac{1}{\beta}} + \frac{b}{3n} \log\frac{1}{\beta}
>         \le
>         \sqrt{\frac{2\sigma^2}{n} \log\frac{1}{\beta}} + \frac{2b}{3n} \log\frac{1}{\beta}
> 		$$
> 		where the last inequality used $\sqrt{a+b} \le \sqrt{a} + \sqrt{b}$ as the reviewer rightly guessed (and similarly for the other tail).
>
> > **Question 2: Do you set $m=K$?**
>
> Yes, all the simulations and experiments in the paper use $m=K$ throughout.
>
> > **Question 3: Are there tradeoffs that make increasing the number of out-canaries $m$ not always beneficial?**
>
> Thank you for pointing out this subtlety – you are right! There is no bias-variance tradeoff associated with the number $m$ of canaries used for the null hypothesis (which we refer to as “test canaries” below) and larger $m$ never hurts.
>
> We show the plots for the synthetic Gaussian setting in **Figure 2 of the rebuttal PDF**. We find a near-monotonic improvement in the lower bound as $m$ grows larger but the improvement quickly saturates at or before $m = 2K$. The bias-variance trade-off w.r.t. $K$ still holds though, irrespective of the value of $m$.
>
> > **Details: Numerical comparison of Bernstein and Wilson intervals in the evaluation**
>
> We refer to **Figure 3 of the rebuttal PDF**. While the Bernstein intervals have slightly worse performance than Wilson intervals as expected, LiDP auditing with Bernstein intervals still leads to improvements over DP auditing with Bernstein intervals.
>
> > **Typos, particularly the missing factor of $\frac{K-1}{K}$**
>
> Great catch, we fixed this and other typos. Thanks!
>
> > **Limitations and suggestions on topics to discuss**
>
> These are great topics worthy of a detailed discussion. We add a brief note here and a more detailed discussion in the paper.
>
> > **Discussion of how the lower bounds depend on the canary design and detection strategy**
>
> We provide a detailed discussion on lower bounds in Appendix A. The lower bound for auditing depends on making the right side of Eq. (1) small. This can be done by canary and rejection set design so that the canary is easy to detect. In other words, $\mathbb{P}(\mathcal{A}(D_1) \in R)$ is large and  $\mathbb{P}(\mathcal{A}(D_0) \in R)$ is small. Much previous work has focused on this (particularly on designing the canary); we review this literature in Appendix A.1. On the other hand, our work focuses on improving the statistical dependence on the number of trials (as reviewed in Appendix A.2).
>
> > **Size and variety of models**
>
> Our analysis in Figures 3 & 4 (and Figures 9 & 11 in the appendix) indicates that random gradient canaries in particular would scale well to larger models. The current bottleneck to larger-scale adoption of rigorous privacy auditing is the prohibitive cost of training multiple models (see also the size of datasets/models in the previous literature [[1](https://arxiv.org/pdf/2006.07709.pdf), [2](https://arxiv.org/pdf/2202.12219.pdf), [3](https://www.computer.org/csdl/proceedings-article/sp/2021/893400b183/1t0x9402gY8), [4](https://openreview.net/pdf?id=AKM3C3tsSx3)]). Our work alleviates this cost to some extent but making privacy auditing feasible at larger scales still requires further research.

---

> > ### Author Response · Authors · 2023-08-21
> > **Thank you for your review! Could you please check the rebuttal?**
> >
> > Thank you again for your thorough review! As the discussion period draws to a close, could you please take a look at the rebuttal and make sure that we have answered all your questions? Thank you!

---

### Author Rebuttal · Authors · 2023-08-09

We thank the reviewers for their detailed and thoughtful comments. We are delighted that the reviewer appreciated the novelty, principled formal analyses, and potential impact of our work.

We are excited that the reviewers recognize that our work "solves a clearly stated problem known in the privacy auditing community" with "innovative and novel tools" that are "compatible with future progress in strengthening canaries." We are pleased that the reviewers appreciate the "first formal analysis of previously used heuristics" and "technically solid theoretical foundations" that are supported by "the necessary experiments."

Below, we respond to each reviewer individually. All additional numerical comparisons requested by the reviewers can be found in the attached PDF. We would be happy to answer any further questions or comments!

---

### Decision · Program_Chairs · 2023-09-21

**Decision:**

Accept (poster)

**Comment:**

This paper proposes a new privacy auditing approach that allows the use of multiple canary samples when auditing a model, thereby improving the efficiency of existing privacy auditing. The approach carefully adapts the hypothesis testing machinery to account for the correlations introduced by the use of multiple canaries. Interestingly, it is agnostic to the way the canaries are generated and can thus be combined with past and future canary generation approaches.

The author response helped to clarify some of the concerns raised by one reviewer, who ended up raising his/her score. In the end, there was a consensus for accepting this paper. Therefore, I recommend acceptance.